

# Unusual chlorine partitioning in the 2015/16 Arctic winter lowermost stratosphere: Observations and simulations

Sören Johansson[1], Michelle L. Santee[2], Jens-Uwe Grooß[3], Michael Höpfner[1], Marleen Braun[1], Felix Friedl-Vallon[1], Farahnaz Khosrawi[1], Oliver Kirner[4], Erik Kretschmer[1], Hermann Oelhaf[1], Johannes Orphal[1], Björn-Martin Sinnhuber[1], Ines Tritscher[3], Jörn Ungermann[3], Kaley A. Walker[5], and Wolfgang Woiwode[1]

[1]Institute of Meteorology and Climate Research, Karlsruhe Institute of Technology, Karlsruhe, Germany
[2]Jet Propulsion Laboratory, California Institute of Technology, Pasadena, California, USA
[3]Institute of Energy and Climate Research - Stratosphere (IEK-7), Forschungszentrum Jülich, Jülich, Germany
[4]Steinbuch Centre for Computing, Karlsruhe Institute of Technology, Karlsruhe, Germany
[5]Department of Physics, University of Toronto, Toronto, Ontario, Canada

**Correspondence:** S. Johansson (soeren.johansson@kit.edu)

**Abstract.** The Arctic winter 2015/16 was characterized by cold stratospheric temperatures. Here we present a comprehensive view of the temporal evolution of chlorine in the lowermost stratosphere (LMS) over the course of this winter. We utilize two-dimensional vertical cross sections of ozone ($O_3$) and chlorine nitrate ($ClONO_2$), measured by the airborne limb-imager GLORIA (Gimballed Limb Observer for Radiance Imaging of the Atmosphere) during the POLSTRACC/ GW-LCYCLE II/ GWEX/ SALSA campaigns, to investigate in detail the tropopause region. Observations from three long-distance flights in January, February and March 2016 are discussed. $ClONO_2$ volume mixing ratios up to 1100 pptv were measured at 380 K potential temperature in mesoscale structures. Similar mesoscale structures are also visible in $O_3$ measurements. Both trace gas measurements are applied to evaluate simulation results from the chemistry transport model CLaMS (Chemical Lagrangian Model of the Stratosphere) and the chemistry climate model EMAC (ECHAM5/MESSy Atmospheric Chemistry). These comparisons show agreement within the expected performance of these models. Satellite measurements from Aura/MLS (Microwave Limb Sounder) and SCISAT/ACE-FTS (Atmospheric Chemistry Experiment - Fourier Transform Spectrometer) provide an overview over the whole winter and information about the stratospheric situation above flight altitude. Time series of these satellite measurements reveal unusually low hydrochloric acid (HCl) and $ClONO_2$ at 380 K from the beginning of January to the end of February 2016, while chlorine monoxide (ClO) is strongly enhanced. In March 2016, unusually rapid chlorine deactivation into HCl is observed instead of deactivation into $ClONO_2$, the more typical pathway for deactivation in the Arctic. Chlorine deactivation observed in the satellite time series is well reproduced by CLaMS. Sensitivity simulations with CLaMS demonstrate the influence of low abundances of $O_3$ and reactive nitrogen ($NO_y$) due to ozone depletion and sedimentation of $NO_y$-containing particles, respectively. On the basis of the different altitude and time ranges of these effects, we conclude that the substantial chlorine deactivation into HCl at 380 K arose as a result of very low ozone abundances together with low temperatures. Additionally, CLaMS estimates ozone depletion of at least 0.4 ppmv at 380 K and 1.75 ppmv at 490 K, which is comparable to other extremely cold Arctic winters. We have used CLaMS trajectories to analyze the history of enhanced



ClONO$_2$ measured by GLORIA. In February, most of the enhanced ClONO$_2$ is traced back to chlorine deactivation that had occurred within the past few days prior to the GLORIA measurement. In March, after the final warming, air masses in which chlorine has previously been deactivated into ClONO$_2$ have been transported in the remnants of the polar vortex towards the location of measurement for at least 11 days.

## 1  Introduction

The chemical reactions and processes that end catalytic ozone (O$_3$) depletion by chlorinated substances in the stratosphere are well understood and explained (Solomon, 1999; von Clarmann, 2013, and references therein). Deactivation into chlorine nitrate (ClONO$_2$) usually is the dominant process for removal of active chlorine in the Arctic winter stratosphere. This is in contrast to the Antarctic, where this process is hindered by the low availability of reactive nitrogen (NO$_y$), which is caused by strong denitrification (Solomon, 1999; WMO, 2007, and references therein). The usual temporal evolution of the chlorine reservoirs in the end of Arctic winters is a sharp increase of ClONO$_2$ followed by a slow increase of hydrochloric acid (HCl; see, e.g., WMO, 2007, Figure 4-10). However, there are exceptions to this general picture. For extraordinarily cold Arctic winters, enhanced chlorine activation and atypically strong chlorine deactivation into HCl have been reported (Santee et al., 2008b; Manney et al., 2011). This preferential deactivation into HCl can be caused by the lack of nitrogen dioxide (NO$_2$) due to strong denitrification, which is usually observed in the Antarctic (e.g., Fahey et al., 1990; Santee et al., 1998) but also occasionally in the Arctic (e.g., Waibel et al., 1999; Santee et al., 2000; Grooß et al., 2005). Additionally, cold temperatures and low O$_3$ abundances can favor chlorine deactivation into HCl (Prather and Jaffe, 1990; Douglass et al., 1995; Grooß et al., 1997, 2011; Mickley et al., 1997). Douglass and Kawa (1999) also demonstrated that even small decreases in O$_3$ together with low stratospheric temperatures can shift chlorine deactivation towards HCl. In this work, we use sensitivity model simulations to quantify these influences on chlorine deactivation for different altitude and time ranges.

Measurements of chlorine species have been performed from space-borne instruments for the chlorine reservoirs ClONO$_2$ (e.g., Zander et al., 1986; Roche et al., 1994; Höpfner et al., 2004; Nakajima et al., 2006; Wolff et al., 2008) and HCl (e.g., Beaver and Russell, 1998; Mahieu et al., 2008; Froidevaux et al., 2008b), and active chlorine such as chlorine monoxide (ClO) (e.g., Waters et al., 1993; Glatthor et al., 2004; Urban et al., 2005; Santee et al., 2008a). These satellite measurements can provide global coverage over a long period of time. These observations have been used to investigate for example chlorine partitioning (e.g., Dessler et al., 1995; Dufour et al., 2006; Santee et al., 2008b), and for model evaluation (e.g., Andersson et al., 2016; Grooß et al., 2018). We utilize time series of Aura/MLS (Microwave Limb Sounder) and SCISAT/ACE-FTS (Atmospheric Chemistry Experiment - Fourier Transform Spectrometer) to compare the Arctic winter 2015/16 with other Arctic winters, and to study the temporal evolution of relevant trace gases in the lowermost stratosphere (LMS) over the course of the winter. In the LMS region, vertical and horizontal resolutions of current satellite missions are limited, which makes these measurements difficult to use for mesoscale case studies. Balloon and airborne in-situ instruments provide precise and spatially highly resolved measurements at the location of the balloon or aircraft, which enables process studies (e.g., Lelieveld et al., 1999; von Hobe et al., 2013) and studies of chlorine partitioning at flight altitude (e.g., Jurkat et al., 2017) to be performed.



Limb remote sensing measurements from balloon and airborne instruments provide trace gas profiles that have been used to study the diurnal cycle of trace gases (Wetzel et al., 2012), chlorine partitioning (von Clarmann et al., 1995; Wetzel et al., 2015), and process studies (e.g von Hobe et al., 2013, and references therein). Most of these studies focus on altitudes of around 500 K potential temperature, where maximum chlorine activation is typically found. Observations of relevant trace gas species

at high spatial resolution in the polar LMS region are sparse. The LMS region is influenced by dynamical effects, such as transport, mixing, wave propagation, and subsidence (e.g., Gettelman et al., 2011), and shows large variability, which requires spatially highly resolved measurements. We utilize such observations to understand the process of chlorine deactivation in this dynamically complex situation in the LMS, and to validate atmospheric chemistry climate and transport models.

Spatially highly resolved measurements have been performed during the PGS (POLSTRACC: POLar STRAtosphere in a

Changing Climate/GW-LCYCLE II: Gravity Wave Life Cycle Experiment/GWEX: Gravity Wave EXperiment/SALSA: Seasonality of Air mass transport and origin in the Lowermost Stratosphere using the HALO Aircraft) campaign in the Arctic winter 2015/16. This particular winter was characterized by exceptionally cold stratospheric temperatures (Manney and Lawrence, 2016; Matthias et al., 2016), by the presence of Polar Stratospheric Clouds (PSCs) down to altitudes lower than 15 km over a long period of time (Pitts et al., 2018; Voigt et al., 2018), and by strong denitrification (Khosrawi et al., 2017). These con-

ditions allowed for extensive activation of chlorine. Due to the long time period of the campaign (from December 2015 to March 2016), it was possible to probe the upper troposphere and lower stratosphere (UTLS) at different times over the course of the Arctic winter. The long distance flights (up to 8000 km) with the German HALO (High Altitude LOng range) research aircraft enabled the GLORIA (Gimballed Limb Observer for Radiance Imaging of the Atmosphere) instrument to measure highly resolved two dimensional trace gas distributions over large distances.

In this work, we use this unique data set to validate the processes for the chemical transport model CLaMS (Chemical Lagrangian Model of the Stratosphere) and the chemistry climate model EMAC (ECHAM5/MESSy Atmospheric Chemistry) at the lower boundary of the polar vortex. Satellite measurements are utilized to give context to the Arctic winter 2015/16, and to investigate the temporal evolution of chlorine activation and deactivation in the LMS in comparison to CLaMS. Sensitivity simulations are performed to estimate the influence of the availability of $O_3$ and $NO_y$ on chlorine deactivation into HCl or

$ClONO_2$. CLaMS is also used to estimate chemical ozone loss in 2015/16, which has not been reported by other studies. Additionally, we investigate the deactivation of chlorine into $ClONO_2$ in the measured air masses. For this purpose, CLaMS is used to estimate the fraction of measured $ClONO_2$ arising from in situ deactivation in the LMS and the fraction that has been transported downwards over the timescale of several days.

## 2  Data sets and methods

### 2.1  PGS Campaign and GLORIA observations

The aircraft campaigns POLSTRACC, GW-LCYCLE II, GWEX, and SALSA were conducted together as the PGS campaign during the Arctic winter 2015/16 from bases in Oberpfaffenhofen, Germany and Kiruna, Sweden. In total, 18 research flights from 17 December 2015 to 18 March 2016 were performed, covering regions between 80° W - 30° E longitude and 25° N -




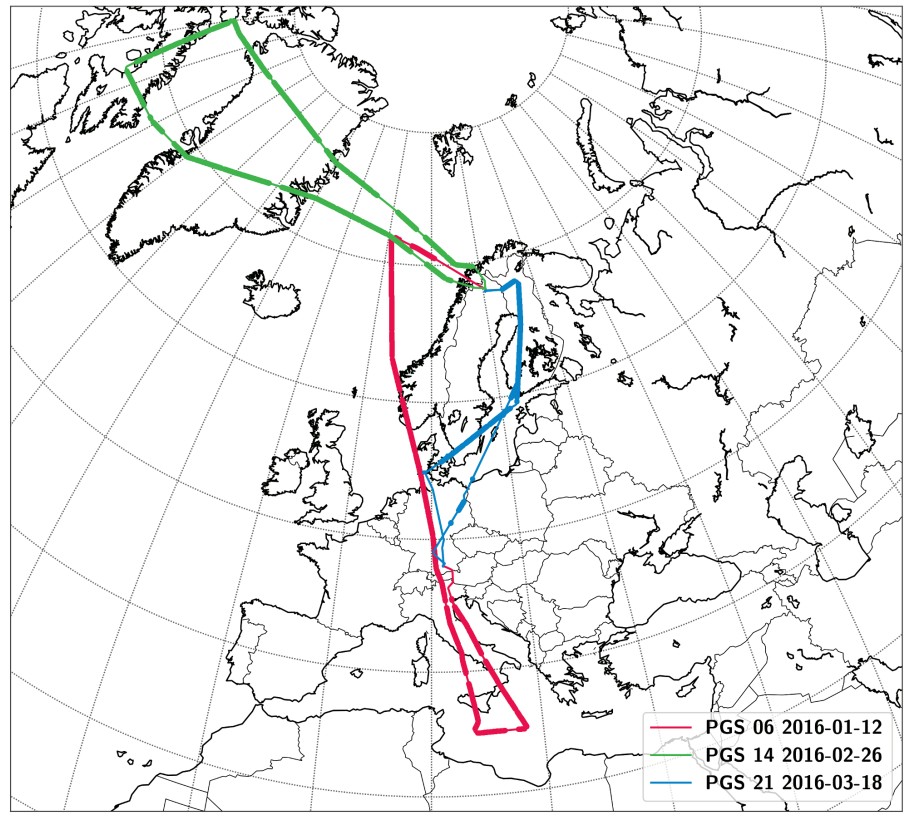

**Figure 1.** Flight paths of flights PGS06 (12 January 2016), PGS14 (26 February 2016) and PGS21 (18 March 2016) on which GLORIA measurements were performed. Aircraft positions with measurements in GLORIA high spectral resolution mode are marked as bold lines.

87° N latitude. Aboard the German research aircraft HALO, nine in-situ and three remote sensing instruments were deployed. The flight paths of flights with GLORIA measurements discussed in this paper are depicted in Fig. 1. GLORIA is an airborne limb imaging Fourier transform spectrometer (Friedl-Vallon et al., 2014) that is mounted below the aircraft and simultaneously records $128 \times 48$ (vertical $\times$ horizontal) interferograms per measurement in across-track limb viewing geometry. These

5    interferograms are radiometrically calibrated using in-flight measurements of two black-bodies. Subsequently the spectra are horizontally averaged within each measurement for noise reduction (Kleinert et al., 2014). From measurements in the high spectral resolution mode (also known as "chemistry mode", with a spectral sampling of $0.0625$ cm$^{-1}$), temperature and trace gases are retrieved with the software KOPRAFIT (Höpfner, 2000). This software uses derivatives of the radiance with respect to the retrieved atmospheric parameter provided by the line-by-line radiative transfer model KOPRA (Karlsruhe Optimized

10    and Precise Radiative transfer Algorithm, Stiller, 2000) to iteratively solve the inverse problem of radiative transport applying the Gauss-Newton algorithm (Rodgers, 2000) with a regularization according to Tikhonov and Arsenin (1977) and Phillips (1962). The data set used in this work contains temperature, $O_3$, $HNO_3$, $ClONO_2$, $H_2O$, and CFC-12. It is described, validated with in-situ measurements, and compared to satellite observations by Johansson et al. (2018). The retrieved temperatures have





a total estimated error of 1 to 2 K. Trace gases of this data set have combined (systematic and random) errors of 10 - 20 %. Vertical resolutions of 400 - 1000 m are achieved. During parts of some of these PGS flights, the GLORIA measurement mode was changed to a reduced spectral and enhanced spatial resolution (also known as "dynamics mode", see Ungermann et al., 2015). These measurements are not discussed in this work and may appear as missing data.

## 2.2 Satellite observations

### 2.2.1 Aura/MLS

The Microwave Limb Sounder (MLS) instrument aboard the NASA Earth Observing System Aura satellite, which was launched in 2004, is a successor to the MLS instrument on the Upper Atmosphere Research Satellite (UARS). It measures thermal radiation in the microwave spectrum (0.1 to 2.5 mm wavelength) in five spectral bands with seven radiometers pointing in the orbital flight direction and vertically scanning the limb of the atmosphere (Waters et al., 2006). With the 705-km altitude near-polar sun-synchronous orbit of the Aura spacecraft, MLS data coverage extends from 82 °S to 82 °N on every orbit, with a vertical limb scan every 165 km along the orbit track. In this work we use the current MLS data version 4.2 of $O_3$, $HNO_3$, HCl, ClO, and $CH_3Cl$. The quality and reliability of the v4 MLS data set are described by Livesey et al. (2018); detailed information on the quality of a previous version (v2.2) of MLS $O_3$, HCl, $HNO_3$, ClO, and $CH_3Cl$ measurements can be found in dedicated evaluation papers (Froidevaux et al., 2008b, a; Santee et al., 2007, 2008a, 2013). In the UTLS region, the vertical resolution and typical estimated error for version 4 trace gases are ≈3 km and ≈10% for $O_3$, ≈4 km and ≈30% for $HNO_3$, 3 km and ≈20-40% for HCl, 3-4.5 km and ≈5-20% for ClO, and 4-5 km and ≈30-45% for $CH_3Cl$. The lowest (i.e., highest pressure) recommended retrieval surface is 261 hPa for $O_3$, 215 hPa for $HNO_3$, and 147 hPa for HCl, ClO and $CH_3Cl$.

### 2.2.2 SCISAT/ACE-FTS

The Atmospheric Chemistry Experiment - Fourier Transform Spectrometer is an infrared limb solar occultation instrument and the main payload of the Canadian SCISAT-1 satellite. The spacecraft was launched in 2003 into a 74° inclination circular orbit at 650 km altitude. In this study, we use $ClONO_2$ from the ACE-FTS version 3.5/3.6 data (Bernath, 2017). The $ClONO_2$ data product has been validated (Sheese et al., 2016) and it has a vertical resolution of 3-4 km and up to 20% estimated error in the UTLS. Due to the solar occultation measurement geometry, no measurements are possible during polar night, and if measurements are possible, the number of measured profiles is significantly lower (up to 30 per day in two latitude "bands") compared to MLS. This limited sampling makes the usage of ACE-FTS daily vortex averaged data difficult, but as shown in various studies the vortex average data quality is good for scientific use (e.g., Dufour et al., 2006; Santee et al., 2008b).

### 2.2.3 CALIPSO/CALIOP

The Cloud-Aerosol Lidar with Orthogonal Polarization (CALIOP) instrument on the CALIPSO (Cloud-Aerosol Lidar and Infrared Pathfinder Satellite Observations) satellite provides measurements of high altitude clouds (PSCs and cirrus) using back-scatter coefficients at 532 nm and 1064 nm of the dual wavelength polarization-sensitive lidar (Winker et al., 2009). The



CALIPSO satellite was launched 2006 and flies in a 98° inclination orbit at 705 km altitude, together (among others) with the Aura satellite in the NASA "A-train" constellation. This allows for nearly coincident measurements of PSCs with Aura/MLS trace gases. A detailed discussion of the CALIOP PSC climatology is given by Pitts et al. (2018).

## 2.3 Model simulations

### 2.3.1 CLaMS

The Chemical Lagrangian Model of the Stratosphere (CLaMS, McKenna et al., 2002b, a; Grooß et al., 2014) is a chemistry transport model (CTM) that has been utilized for a simulation of the chemical composition of the Arctic winter 2015/16 (Grooß et al., 2018). The version "PL8" model run was initialized 1 November 2015 with data from MLS observations of $O_3$, $N_2O$, $H_2O$, and HCl, and over the course of the winter, the dynamics have been specified by wind and temperature fields from the ERA-Interim analysis provided by the ECMWF (European Centre for Medium-Range Weather Forecasts). $CH_3Cl$ has been initialized using a correlation between a 15-day average of CFC-11 simulated by CLaMS and $CH_3Cl$ measured by MLS. This correlation was used instead of CFC-11/$CH_3Cl$ correlations available from ACE-FTS (Brown et al., 2013) because MLS and ACE-FTS show differences in the LMS (Santee et al., 2013), where this study is focused. The model run employs 32 vertical entropy-preserving layers (Konopka et al., 2007) with an altitude-dependent vertical resolution of 400 m at 10 km up to about 800 m between 12 and 24 km altitude and a horizontal resolution of 100 km. The troposphere below 9 km only has a vertical resolution of about 2 km. Recent improvements to the model, such as the influence of galactic cosmic rays (Grooß et al., 2018), are considered in this model run.

For the comparison of CLaMS results to GLORIA and MLS measurements, backward trajectories are calculated from the geolocations of the measurements. At 12:00 UTC, the CLaMS output is spatially interpolated linearly onto these trajectory positions. Chemistry relevant for diurnal variations of target trace gases has been calculated along these trajectories.

The CLaMS sedimentation module offers the possibility to simulate NAT (Grooß et al., 2014) and ice (Tritscher et al., 2018) PSC cloud formation and corresponding denitrification and nitrification as well as dehydration and hydration. Information about the surface area density of ice, NAT, and STS particles per volume of air is available for every CLaMS air parcel. For comparison with the CALIOP PSC areal coverage, lower boundaries for the different surface area densities were defined to discriminate PSCs from background aerosols. In accordance with the CALIOP detection thresholds, PSC thresholds for CLaMS simulations are as follows: 3.3 $\mu m^2$ cm$^{-3}$ for STS droplets (Carslaw et al., 1994), 0.25 $\mu m^2$ cm$^{-3}$ for NAT, and 0.5 $\mu m^2$ cm$^{-3}$ for ice particles. Values exceeding those thresholds are counted as PSCs in those specific composition classes, respectively.

### 2.3.2 EMAC

The ECHAM5/MESSy Atmospheric Chemistry (EMAC, Jöckel et al., 2010) model is an Eulerian chemistry climate model (CCM) that uses the fifth-generation European Centre Hamburg general circulation model (ECHAM5 version 5.3.02, Roeckner et al., 2006) and the second version of the Modular Earth Submodel System (MESSy version 2.52). For the Arctic winter



2015/16 a simulation nudged to the dynamics of the ECMWF operational analyses has been initialized 1 July 2015. This model run was performed with a T106L90MA resolution with a spherical truncation of T106 (which corresponds to a horizontal resolution of approximately $1.125° \times 1.125°$ (latitude $\times$ longitude)) and 90 vertical hybrid pressure levels from the surface up to 0.01 hPa (approx. 80 km). The vertical resolution in the UTLS is about 0.5 km. A comprehensive chemistry set-up with gas-

phase and also heterogeneous reactions on PSCs was included using rate constants mainly from the Jet Propulsion Laboratory (JPL, Sander et al., 2011). A detailed description of this model simulation is given by Khosrawi et al. (2017).

### 2.4    Meteorological analyses and vortex boundary estimation

#### 2.4.1    MERRA2 meteorological reanalysis

The Modern-Era Retrospective analysis for Research and Applications, Version 2 (MERRA2) is the standard meteorological

reanalysis data set of the NASA Global Modeling and Assimilation Office (Gelaro et al., 2017). Global fields of temperature, pressure, potential temperature, and potential vorticity at $0.625° \times 0.5°$ (longitude $\times$ latitude) horizontal resolution and 83 vertical levels are used in this study. Note that both atmospheric models used in this study, CLaMS and EMAC, apply meteorological data from ECMWF rather than MERRA2.

#### 2.4.2    Vortex boundary estimation

The definition of the edge of the polar vortex in the UTLS region is challenging, especially at lower altitudes (Gettelman et al., 2011; Lawrence et al., 2018). In this study, it was decided to use two vortex filters: The first filter calculates the PV determining the edge of the polar vortex according to Nash et al. (1996). The polar vortex at 370 K potential temperature ($\theta$) is shown on maps in order to illustrate its position. The second filter uses scaled potential vorticity (sPV), which is the PV divided by the factor $\partial\theta/\partial p$ (where $p$ denotes the pressure), to exclude the altitude dependency from the PV (Dunkerton and Delisi, 1986;

Manney et al., 1994). The MERRA2 sPV is linearly interpolated onto the measurement geolocations, and measurements with a corresponding sPV $< 1.2 \cdot 10^{-4}$ s$^{-1}$ are filtered out.

### 3    Measurements and model evaluation

In this section, the Arctic winter of 2015/16 is put into climatological context by comparing time series of satellite measurements of the LMS in that year with those from other Arctic winters. A more detailed view of the 2015/16 satellite time series is

then compared to CLaMS simulation results. After that, three exemplary flights from the PGS campaign are discussed in detail regarding GLORIA $O_3$ and $ClONO_2$ VMR (Volume Mixing Ratio) cross sections. These aircraft measurements are then used to validate EMAC and CLaMS simulations.



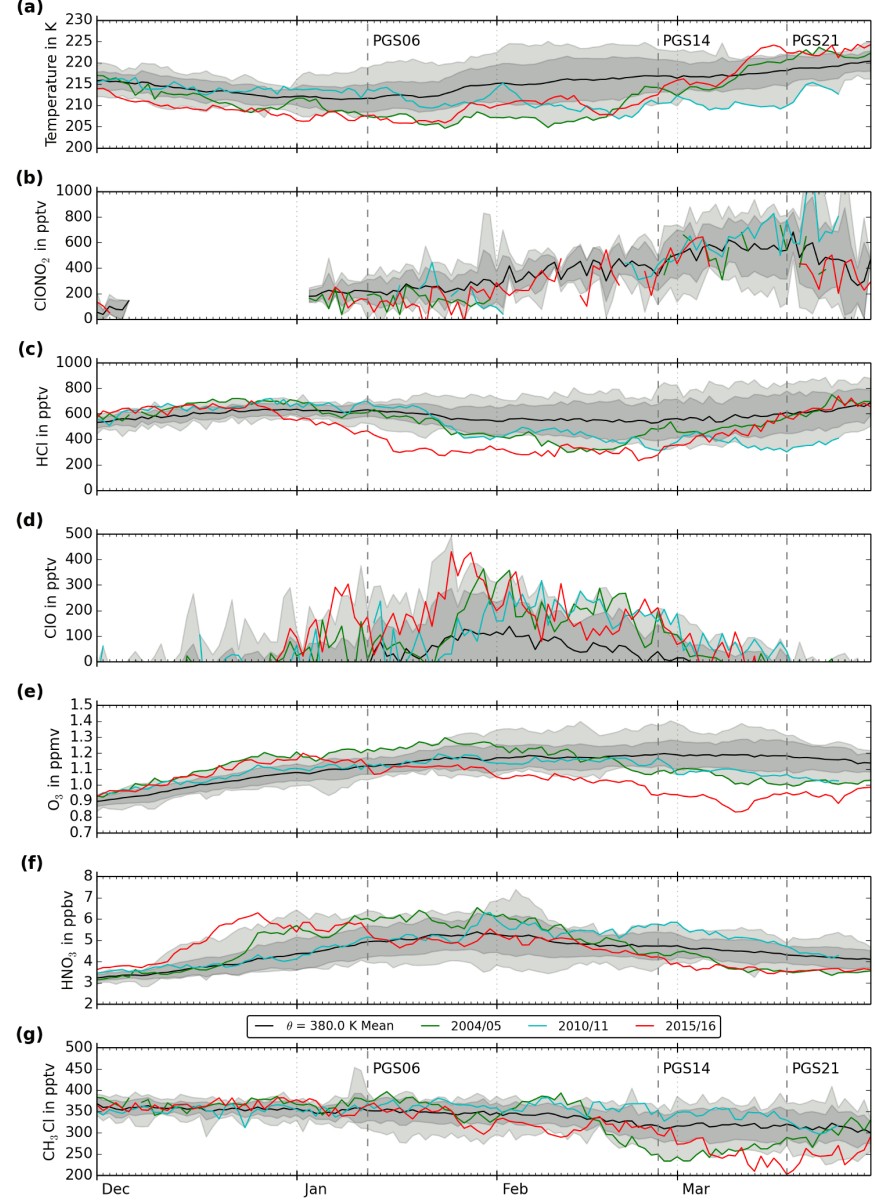

**Figure 2.** Vortex average time series of (a) temperature (MERRA2), and volume mixing ratios of (b) ClONO₂ (ACE-FTS), (c) HCl, (d) daytime ClO, (e) O₃, (f) HNO₃, (g) CH₃Cl (Aura/MLS) for the period 2004-2018 over the Arctic at $\theta$=380K ($\approx$ 15 km in January). The black line marks the average value for each day, the dark shaded area marks the standard deviation of each day between different years and the light shaded area marks minimum/maximum values within the time period. The Arctic winter 2015/16 has been excluded from all of these measures. Mean values for the Arctic winters 2004/05 (green), 2010/11 (cyan), and 2015/16 (red) are marked separately. Days with aircraft measurements discussed in this paper are marked with dashed lines.





## 3.1 Time series

Time series of MERRA2 temperature and MLS and ACE-FTS trace gas data are used to provide context for the aircraft based measurements. These time series are created by averaging all reanalysis temperatures and all measured trace gas profiles (data are vortex filtered according to the sPV criterion defined in Sec. 2.4.2) linearly interpolated to levels of potential temperature

for each day within the polar cap (latitudes > 55 °N). Because of the pronounced diurnal cycle of ClO, all ClO nighttime measurements have been filtered out using the solar zenith angle (threshold of 90°) associated with the measurement. In order to compare the time series of the Arctic winter 2015/16 with other winters, these time series are created for all Arctic winters of the Aura/MLS and ACE-FTS epoch from 2004/05 to 2017/18.

The time series at $\theta$ = 380 K for temperature, ClONO$_2$, HCl, ClO, O$_3$, HNO$_3$, and CH$_3$Cl are shown in Fig. 2. For each

species, the black line marks the mean for each day for all years from 2004 to 2018 (excluding 2015/16), dark grey shading depicts the standard deviation around this mean value and light grey shading marks minimum/maximum values of all time series (excluding 2015/16). The time series for the Arctic winter 2015/16, which is the focus of this study, is presented in red. The Arctic winters 2004/05 (green) and 2010/11 (cyan) are also highlighted. These winters are known for extremely low stratospheric temperatures and have been discussed in detail previously (e.g., Santee et al., 2008b; Manney et al., 2011).

In 2015/16, MERRA2 temperatures at 380 K were near or below the climatological minima through mid-January (Fig. 2a). Then two minor warmings (end of January and mid of February) and the final warming (early March) are visible in the 2015/16 temperature curve. ClONO$_2$ from ACE-FTS measurements (Fig. 2b) exhibits lower abundances in January 2016 compared to the other winters in the ACE-FTS record. Measurements available for February and March show that ClONO$_2$ followed a course in 2016 comparable to or below the multi-year average. The other chlorine reservoir HCl (Fig. 2c) sets a new minimum

of all Aura/MLS time series in January and February for the year 2016. Consistent with the picture from HCl, the MLS ClO (Fig. 2d) is above the average for the whole winter and also establishes new maximum values on many days. O$_3$ in the LMS is slightly above the MLS average (Fig. 2e) in the beginning of the Arctic winter 2015/16, while minimum O$_3$ values are observed from mid February 2016 onward at 380 K. As discussed by Manney and Lawrence (2016), such extremely low O$_3$ has not been observed at higher altitudes (490 K), where the ozone loss was maximum in 2011. HNO$_3$ (Fig. 2f) shows extreme behavior in

2015/16 at $\theta$ = 380 K: In December 2015 HNO$_3$ also exhibits a strong increase, reaching maximum values > 6 ppbv, whereas in March values consistently below 3 ppbv are measured. Methyl chloride (CH$_3$Cl), a largely biogenic trace gas that is well mixed in the troposphere and photolyzed in the stratosphere, is a useful tracer of diabatic descent within the polar vortex. The time series of CH$_3$Cl is illustrated in Fig. 2g: The typically slow decrease of CH$_3$Cl due to diabatic descent is visible in the average VMR, which slowly decreases. During 2015/16, CH$_3$Cl followed a more or less climatological evolution until the

beginning of March, when VMRs dropped rapidly and became highly variable with the onset of the major final warming.

Manney and Lawrence (2016) showed the extraordinary nature of the 2015/16 Arctic winter higher in the stratosphere (at 490 K). The contextual information from satellite observations in Fig. 2 illustrates how exceptional this winter also was in the LMS, where the PGS measurements were performed. At 380 K, the decrease in HCl and increase in ClO, indicative of substantial chlorine activation in the LMS, started earlier and reached more extreme levels in 2015/16 compared to any other

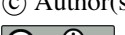



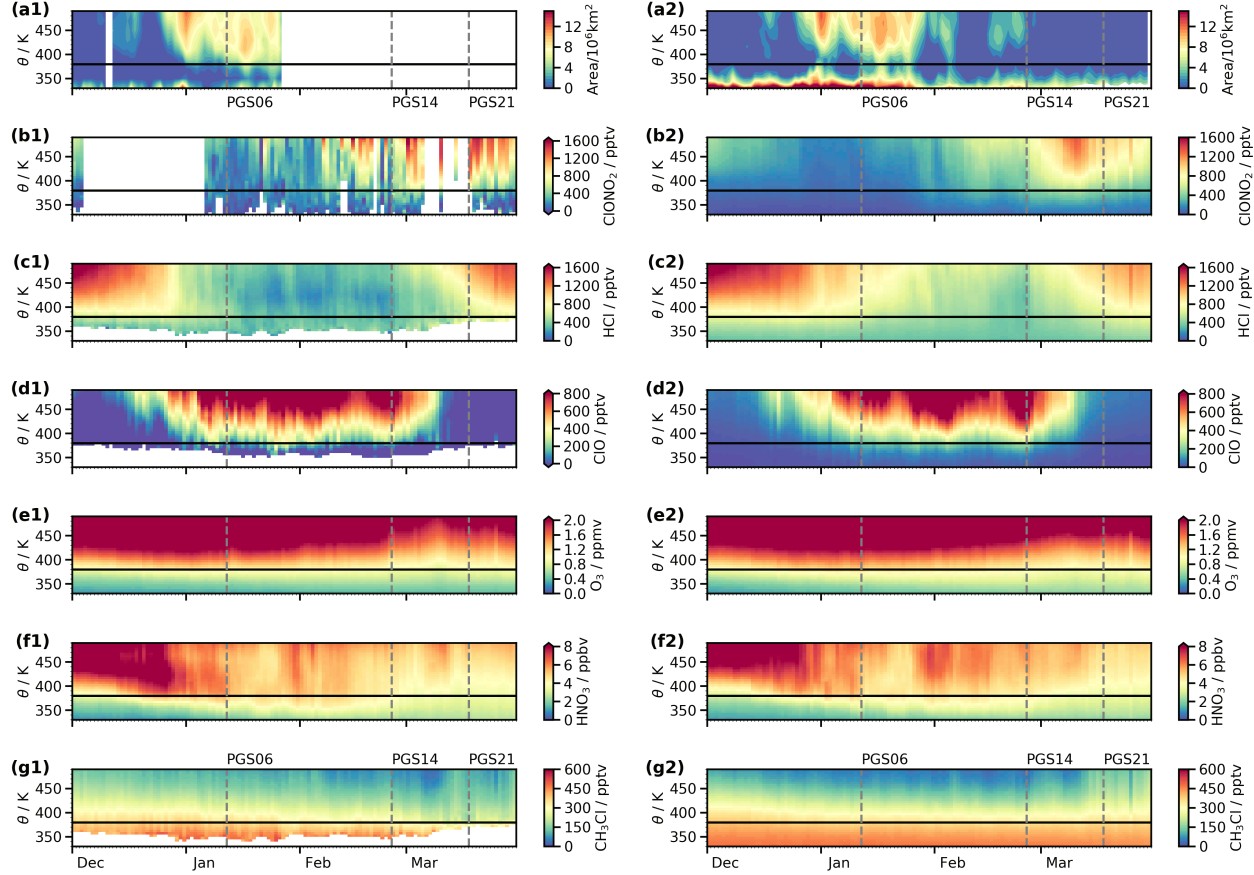

**Figure 3.** Comparison of MLS, ACE-FTS, and CALIOP time series (left) with CLaMS simulation results (right) for (a) PSC areas, (b) ClONO$_2$, (c) HCl, (d) daytime ClO, (e) O$_3$, (f) HNO$_3$, and (g) CH$_3$Cl. The $\theta$ = 380 K altitude is marked with a black line, and days with aircraft measurements discussed in this paper are marked with dashed lines.

year in the Aura/MLS record. The unusually strong chlorine activation at this level is also evident in the ClONO$_2$ abundances measured by ACE-FTS, which lie below the climatological mean in January 2016. Substantial chlorine activation in the Arctic LMS has rarely been reported previously (e.g., Santee et al., 2011). The diabatic descent at 380 K in 2015/16 essentially followed the climatological mean until the final warming in March, as seen in CH$_3$Cl. Thus, the degree of replenishment of O$_3$
5  at 380 K by diabatic descent was not unusual throughout most of the ozone destruction period in 2015/16. For that reason, the unusually low O$_3$ abundances at 380 K indicate chemical ozone loss.

## 3.2 Comparison of satellite measurements to model simulations

For a comparison with the measured Arctic winter 2015/16 time series, the CLaMS simulations have been linearly interpolated to Aura/MLS geolocations (see Sec. 2.3.1), which have been used to calculate vortex averaged profiles in the same manner as



described in Sec. 3.1. In order to get a more complete picture of the CLaMS simulation, modeled ClONO$_2$ (which is measured by ACE-FTS) is also interpolated to these MLS geolocations. The comparison of the time series in the altitude range $\theta$ = 330 - 490 K is illustrated in Fig. 3. Fig. 3a presents daily PSC area time series until the end of January 2016, when instrumental problems forced CALIOP to suspend measurements. The data product shown does not discriminate between cirrus clouds

and PSCs in the UTLS region. Chlorine activation is also possible on cirrus clouds (e.g., Borrmann et al., 1996), and thus this panel indicates the area of potential heterogeneous chlorine activation. These areas increase considerably at 490 K at the end of December 2015 and were also observed at potential temperatures as low as 380 K in the beginning of January 2016. The CLaMS simulation reveals the same temporal and spatial distribution of PSCs as measured by CALIOP, but with slightly larger maximum PSC areas simulated than measured. For the rest of the winter (after CALIOP stopped measurements), CLaMS

shows a rapid decrease of PSC area until the beginning of February and a short period with a small area of PSC occurrences at the end of February. ClONO$_2$ (Fig. 3b) displays enhancements of > 1000 pptv at altitudes $\theta$ > 380 K in March in the simulation and measurements. In the ACE-FTS measurements, these enhancements are also visible starting from the middle of February, whereas the CLaMS simulation indicates weaker maxima of 500 pptv at that time. A direct comparison of these plots is difficult due to the sparse sampling of the ACE-FTS measurements. HCl (Fig. 3c) decreases from 1600 pptv starting in

December in MLS observations and in simulated data. This decrease advances faster to lower minimum values of 200 pptv in the measurements compared to the simulation, which shows a slower decrease to minimum values of 400 pptv. The increase of HCl, starting in the beginning of March, exhibits the same temporal evolution, while maximum values of 1400 pptv are observed and 1200 pptv are simulated. Measured and modeled ClO (Fig. 3d) are enhanced in the same time periods and altitude levels, although in the CLaMS data the enhancement towards the end of December shows lower absolute values

(400 pptv at 490 K) compared to MLS (800 pptv at 490 K). O$_3$ (Fig. 3e) displays very similar curtains over the course of the winter, although towards the end of the winter lower O$_3$ values are observed than simulated at altitudes of $\theta \approx$ 450 K. HNO$_3$ (Fig. 3f) simulated by CLaMS compares well with the corresponding MLS measurement. The major difference is observed at the end of December, when HNO$_3$ values > 8 ppbv are measured at altitudes of $\theta \approx$ 400 K, but only 6 ppbv are simulated at this time and altitude. Fig. 3g presents measured and modeled CH$_3$Cl with VMRs up to 500 pptv in the troposphere and VMRs

measured as low as 50 pptv and simulated as low as 25 pptv in the stratosphere.

For a more quantitative comparison, cuts along $\theta$ = 380 K and $\theta$ = 490 K are presented in Fig. 4. The lower altitude curves are provided as an extension of Fig. 2 to give context to the aircraft measurements in the following sections. The slice at $\theta$ = 490 K is meant as a connection to previous discussions of this winter (Manney and Lawrence, 2016) and to compare to other extreme winters discussed at this altitude level (Santee et al., 2008b; Manney et al., 2011). In addition, showing both levels

illuminates the differences between the LMS and the bulk of the stratosphere above. The chlorine species (Fig. 4a-b), HNO$_3$ (4c-d, left axis), and O$_3$ (4c-d, right axis) show in detail the overall agreement and specific differences between measurement and simulation, which have already been described for Fig. 3. At $\theta$ = 380 K, large discrepancies between model and observation are noted for HCl ($\Delta$vmr$_{max}$ = 200 pptv) and HNO$_3$ ($\Delta$vmr$_{max}$ = 1 ppbv) until the middle of February, while at $\theta$ = 490 K, discrepancies are visible for HCl ($\Delta$vmr$_{max}$ = 400 pptv) starting from the beginning of the winter, and for O$_3$ ($\Delta$vmr$_{max}$ =

750 ppbv) starting from January. CH$_3$Cl (Fig. 4e,f) shows agreement at 380 K until the end of January 2016, with slowly





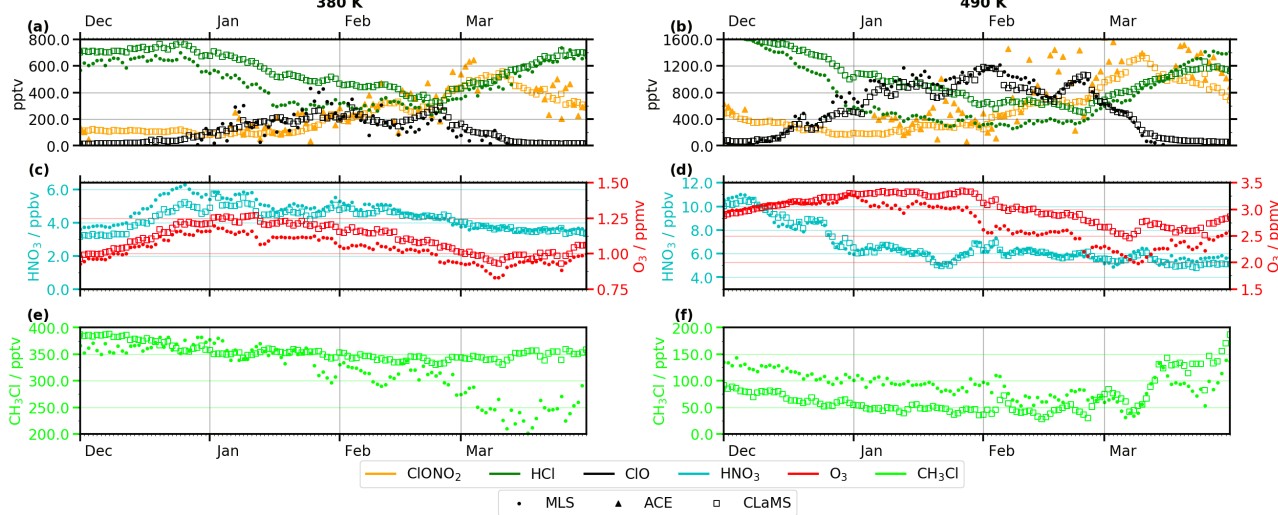

**Figure 4.** Time series of vortex averaged trace gases from satellite measurements and CLaMS simulation at 380 K (a,c,e) and 490 K (b,d,f). CLaMS data are shown as open squares, MLS as filled points and ACE-FTS as triangles. The evolution of chlorine species ($ClONO_2$: orange, HCl: green, ClO: black) is illustrated in the first row (a,b), $HNO_3$ (cyan) and $O_3$ (red) in the second row (c,d), and $CH_3Cl$ (light green) in the third row (e,f).

decreasing VMRs between 380 pptv and 340 pptv. Starting at the end of January, the measured $CH_3Cl$ VMR decreases to values as low as 200 pptv in March, while the simulated VMR remains at about 340 pptv. At 490 K, a difference of 50 pptv between simulated and measured $CH_3Cl$ is visible, with simulated VMRs lower than those measured until February, when the measured $CH_3Cl$ VMRs approach the simulated values. Modeled and measured $CH_3Cl$ then agree until the middle of March,

after which the observed variations are not fully captured by the simulation.

    In general, CLaMS succeeds in reproducing chlorine deactivation of the Arctic winter 2015/16, which has been identified to be unusual by the comparison to time series of other years in the MLS record: The deactivation of chlorine in the Arctic LMS typically starts with a decrease of ClO and an increase of $ClONO_2$, followed by a slow increase of HCl until the equilibrium between the reservoirs is reestablished (Solomon, 1999). The 2015/16 time series of $ClONO_2$ is mostly at or below the 2004-

2018 average in February and March, and an increase of HCl from exceptionally low values of 200 pptv at the end of February 2016 to the 2004-2018 average values of 600 pptv at 380 K is observed in the middle of March 2016 (Figs. 2 and 4).

    CALIOP PSC area measurements agree with PSC areas simulated by CLaMS for the time CALIOP was able to collect measurements. ACE-FTS measurements and CLaMS simulation of $ClONO_2$ show agreement within the limited ACE-FTS sampling, which influences the daily mean profiles. The HCl time series exhibit differences similar to those discussed in detail

by Grooß et al. (2018): In the beginning of the winter, activation of HCl is not simulated to the extent it is observed by the satellite instrument. This lack of chlorine activation is also visible in ClO, which indicates lower VMRs in the model in the beginning of the winter. Another result of the reduced chlorine activation of CLaMS is the overestimation of simulated $O_3$



compared to MLS observations, which is in particular visible at 490 K towards the end of the winter. $HNO_3$ shows agreement between MLS and CLaMS, but in the beginning of the winter lower VMRs are simulated than observed at 380 K. This disagreement is considered to result from an underestimation by CLaMS of re-nitrification at this level from the sedimentation of $HNO_3$ containing PSC particles from above. It is known that denitrification and re-nitrification are difficult to simulate in

the LMS (Braun et al., 2019). Comparisons of $CH_3Cl$ show agreement between measurement and simulation for the beginning of the winter at 380 K, but starting in January measured $CH_3Cl$ decreases notably, while the simulated $CH_3Cl$ remains almost constant. These differences indicate that diabatic descent is too weak in the model at 380 K. At 490 K, the persistent 50 pptv model-measurement discrepancy suggests that the $CH_3Cl$ VMRs used to initialize the simulation (see Sec. 2.3.1) were too low, which does not allow for further conclusions at this altitude level.

### 3.3 Aircraft measurements

In addition to the overview of the Arctic winter 2015/16 chemical composition of the LMS using satellite observations, the airborne GLORIA measurements of $O_3$ and $ClONO_2$ aim to give detailed insights of the lowermost part of the polar vortex. Out of 14 scientific flights, 3 flights with particularly interesting trace gas distributions at different stages of the winter are discussed in this section. Measurements from other flights are provided in the supplement of Johansson et al. (2018).

### 3.3.1 Flight on 12 January 2016 (PGS06)

The flight on 12 January 2016 (PGS06, Fig. 5) during the early/mid winter was the transfer flight from the campaign base Oberpfaffenhofen (Germany) to Kiruna (Sweden) via southern Italy (way points "A" and "B", marked in Fig. 5). As shown in Fig. 5a, the polar vortex (estimated by regions that are not marked with a shadow) extended over central Europe to the Arctic and Siberia. It can be seen from the higher MERRA2 potential temperature that air masses at the typical flight altitude of 13 km

had subsided over southern France and northern Italy (close to way point "C"), where a tropopause fold was present along the polar front jet stream on the day of the flight (Woiwode et al., 2018). Towards the end of the flight (after way point "D"), high altitude clouds were observed along the GLORIA line of sight, and no retrievals were possible.

Two dimensional trace gas distributions of $O_3$ (Fig. 5b) and $ClONO_2$ (Fig. 5c) along the flight path are shown as a function of time and $\theta$ for PGS06 and in the following sections for PGS14 and PGS21. In order to compensate for dynamical features in the

atmosphere, the trace gas cross sections are linearly interpolated on potential temperature levels. Additionally, the MERRA2 potential vorticity (2 and 4 PVU (potential vorticity units)) along the measurement geolocations are plotted as magenta lines to identify the dynamical tropopause. Way points, marked by capital letters and dashed lines, help to arrange these curtain plots on the map. Trace gas retrievals are possible between cloud top and flight altitude and regions outside this range are not shown. In addition, for time periods used for calibration measurements, refuel stops (only on flight PGS21), and different measurement

modes (only on flight PGS21) no retrieval results are available.

The measured $O_3$ concentrations reveal enhanced values up to 1200 ppbv at way point "C" at potential temperatures of 370 K. Below this maximum, small-scale structures of $\sim$800 ppbv are visible. Ozone values are low in tropospheric air masses over Italy (near way points "A" and "B"), but during the rest of the flight for the most part VMRs of $\approx$500 ppbv are observed





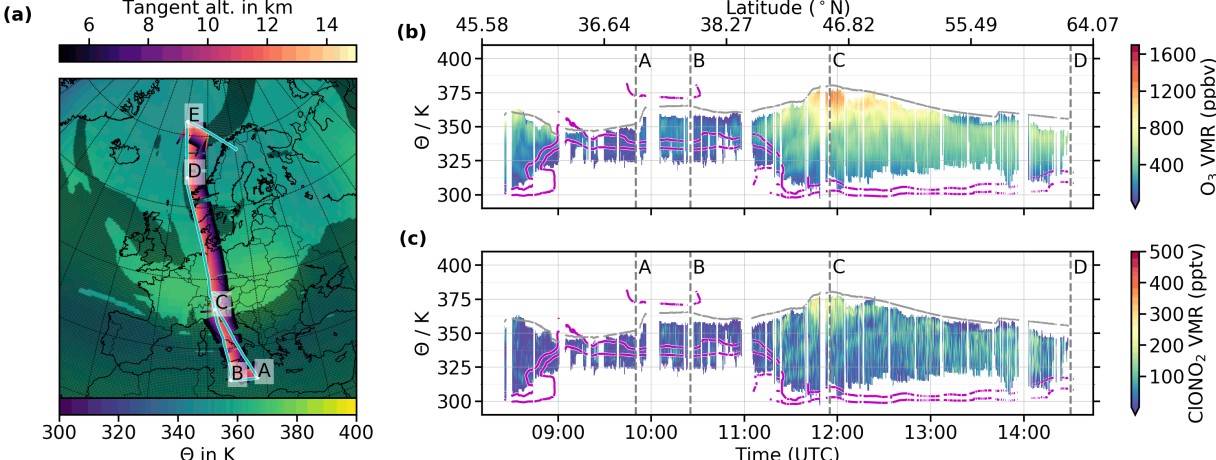

**Figure 5.** Flight PGS06 (12 January 2016): (a) Flight path (cyan) on a map with MERRA2 potential temperatures (lower colorbar) at typical HALO cruise altitude of 13 km. Regions outside the polar vortex (according to Nash et al. (1996) at $\theta$ = 370 K) are marked with a grey shadow. Way points are marked with capital letters. The tangent altitudes of GLORIA measurements are shown in the upper colorbar. Right panels: GLORIA cross section of (b) $O_3$ and (c) $ClONO_2$ (using GLORIA potential temperature). The flight level (approximated using MERRA2 potential temperature) is marked with a grey line and white spaces mark regions without data. The MERRA2 potential vorticities of 2 and 4 PVU are marked with magenta lines and way points are marked with grey vertical dashed lines.

at potential temperatures between 310 and 350 K. $ClONO_2$ shows maximum values up to 250 pptv close to the maximum $O_3$ VMR at way point "C", where the highest potential temperature levels were reached. In the second part of this flight (after 11:30 UTC), small $ClONO_2$ enhancements (< 200 pptv) are visible in small-scale horizontal structures at 310-350 K potential temperature. Overall the GLORIA observations show a long north-south transect through the Arctic LMS in mid-January 2016,
5    with subsided ozone-rich air masses and first indications of chlorine deactivation into $ClONO_2$.

### 3.3.2   Flight on 26 February 2016 (PGS14)

In the middle of the Arctic winter on 26 February 2016, flight PGS14 was realized as shown in Fig. 6a. From the campaign base in Kiruna, the flight headed towards the northern part of Greenland (way point "A"), continued until Baffin Bay (way point "B"), and turned at way point "C" to change direction towards Kiruna. At a typical flight altitude of 13 km, the substantially
10   subsided air masses are evident in the higher potential temperatures characterizing the portion of the flight track near way point "B". The vortex criterion of Nash et al. (1996) at $\theta$ = 370 K also shows that most of the flight path was within the polar vortex.

The cross sections of $O_3$ (Fig. 6b) and $ClONO_2$ (Fig. 6c) are presented in the same manner as for flight PGS06. Between way points "A" and "B", maximum $O_3$ values of 1600 ppbv are measured at $\theta$ = 390 K, and below small-scale structures are visible at $\theta$ = 340 K (in the vicinity of way point "C"). $ClONO_2$ developed a local maximum of 600 pptv below flight altitude


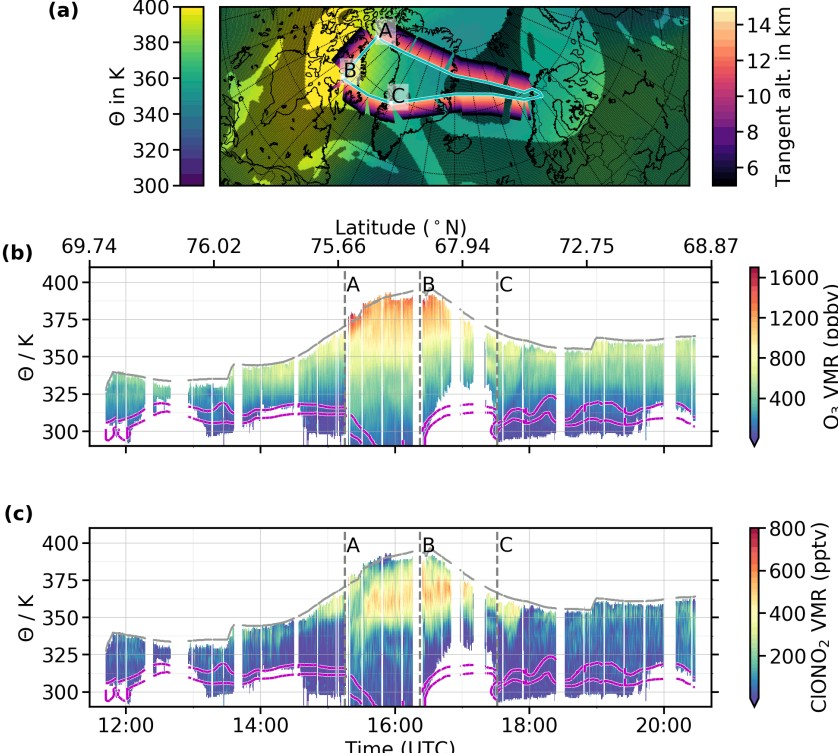

**Figure 6.** As in Fig. 5, but for Flight PGS14 (26 February 2016).

at $\theta = 360$ K around way points "A" and "B". The maximum that becomes visible just before way point "A" is discontinuous, reforming at slightly lower altitudes along the flight path (diagonal local minimum feature in Fig. 6c).

### 3.3.3 Flight on 18 March 2016 (PGS21)

The flight path of the late winter flight on 18 March 2016 (PGS21) is shown in Fig. 7a on a map with MERRA2 potential

5 temperature at the typical flight altitude of 13 km. This flight started at the campaign base in Oberpfaffenhofen and headed towards Denmark (way point "A"), where the GLORIA measurement mode was changed to the "chemistry mode". Then the flight course followed the Baltic sea northeastwards until it reached remains of the late winter polar vortex at way point "B". Inside this region of high potential temperatures, the HALO aircraft continued northward until a refueling stop in Kiruna (shortly after way point "C"). On its way back to Oberpfaffenhofen, HALO took a similar flight path over the Baltic sea until

10 the measurement mode of GLORIA was changed (way point "D").

Two dimensional cross sections of $O_3$ and $ClONO_2$ are shown in Fig. 7b,c. Between way points "B" and "C", maximum $O_3$ values of 1600 ppbv were observed and filamentary structures are visible down to altitudes of $\theta = 340$ K. Between way points "A" and "B", these filaments are visible close to flight altitude at $\theta = 360$ K. For $ClONO_2$, enhanced values up to 1100 pptv are





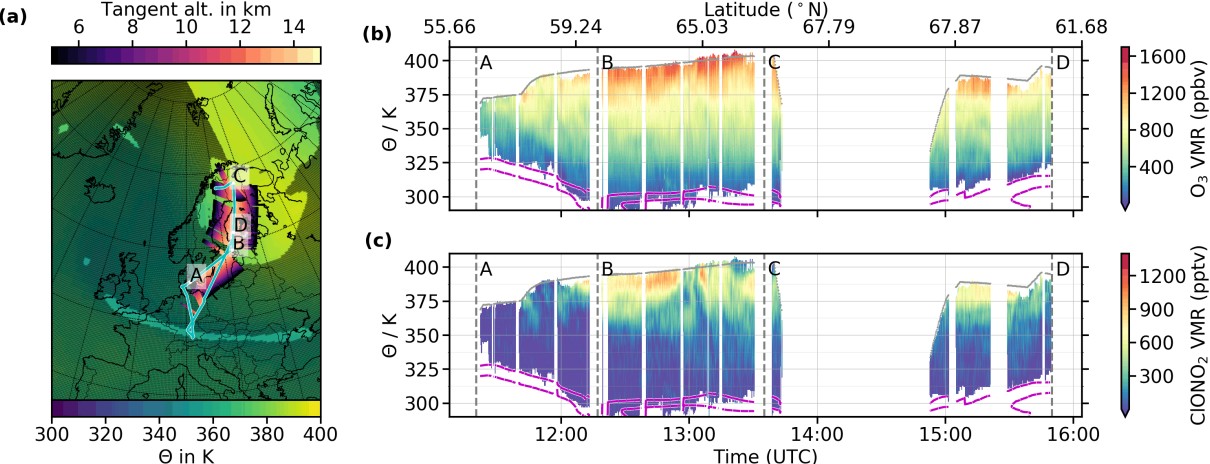

**Figure 7.** As in Fig. 5, but for Flight PGS21 (18 March 2016).

measured at $\theta = 380$ K altitude. ClONO$_2$ as high as 500 pptv is measured at altitudes as low as $\theta = 350$ K. Between way points "B" and "C", where consistently high O$_3$ VMR is observed at $\theta = 370$ K, strong horizontal fluctuations in ClONO$_2$ are visible. These filaments in ClONO$_2$ are likely to be connected to the availability of NO$_2$. Structures of HNO$_3$ (which is photolyzed to NO$_2$) and ClONO$_2$ in the collocated GLORIA measurements reveal similar shapes (see Johansson et al., 2018, supplement).

## 3.4 Comparison of aircraft measurements to model simulations

The CLaMS results have been compared to various observations of Arctic winters, which led to important improvements of this model (e.g., Grooß et al., 2014, 2018; Tritscher et al., 2018). Also for the EMAC model, comparisons with different observations of Arctic winters have been performed (e.g., Khosrawi et al., 2017, 2018). Still, those previous comparisons focus on the stratosphere and comparisons in the LMS are only marginally discussed. Complex dynamical situations, which may occur in the LMS, are challenging for atmospheric modeling. Thus, comparisons in this altitude region are beneficial to benchmark the performance of these models.

The GLORIA cross sections of O$_3$ and ClONO$_2$ are compared to the CLaMS and EMAC model results in Figs. 8 and 9. CLaMS output has been interpolated to GLORIA geolocations as described in Sec. 2.3.1. The globally available EMAC data have been interpolated linearly to the GLORIA tangent point geolocations. ECMWF operational analysis pressure, linearly interpolated to the GLORIA tangent point geolocations, is used to assign the EMAC model vertical levels (provided on pressure levels) to the GLORIA retrieval altitudes of the measurements. EMAC model output was provided every five hours.

For O$_3$ on flight PGS06 (Fig. 8a,d,g), the low tropospheric VMR values that can be found above Italy (9:00 - 11:00 UTC) are as well reproduced by CLaMS (compared to GLORIA) as enhanced values above central and northern Europe. Maximum modeled values at 12:00 UTC of 1200 ppbv are slightly lower than measured (1400 ppbv) but agree within the total estimated




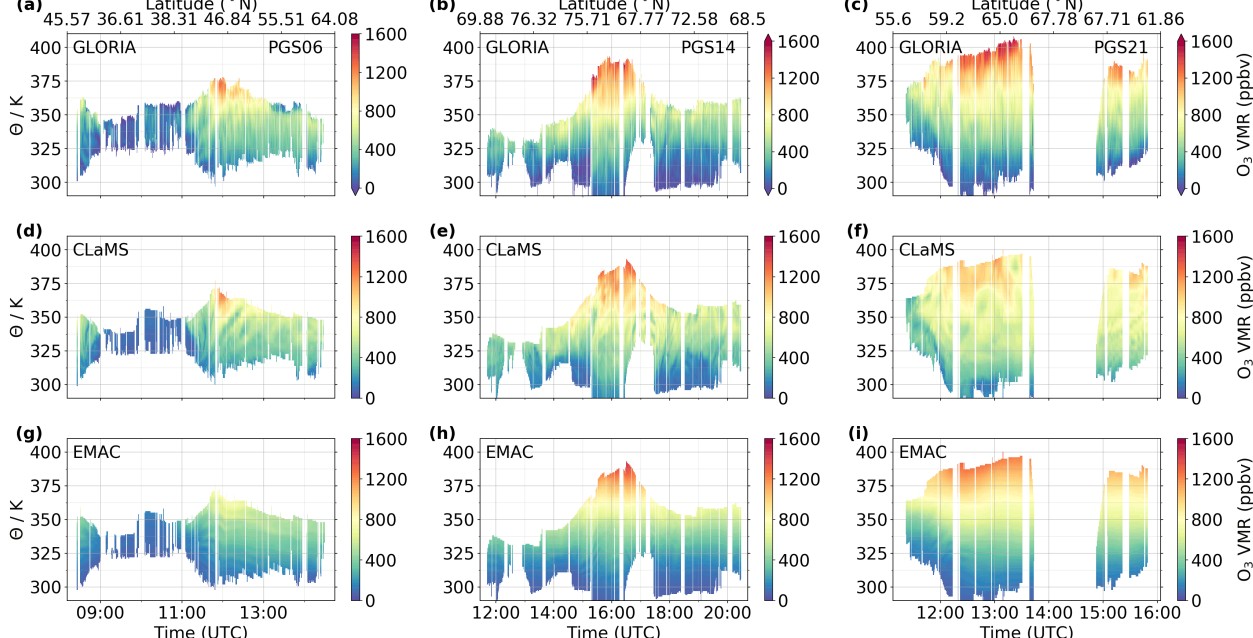

**Figure 8.** Comparison of GLORIA measured (first row, repeated from Figs. 5-7 with a different color bar) to CLaMS (second row) and EMAC (third row) simulated $O_3$ cross sections for flights PGS06 (12 January 2016, first column), PGS14 (26 February 2016, second column) and PGS21 (18 March 2016, third column). Colorbars with top/bottom arrowheads indicate data greater/smaller than the colorbar range.

error of the instrument. A local minimum of $O_3$ observed at flight altitude after 13:00 UTC is not present in CLaMS or EMAC, but is suspected to be influenced by a slight degradation of the GLORIA measurements due to PSCs (see in-situ comparisons in Johansson et al., 2018, supplement). The comparison with EMAC shows that the overall structure of measured $O_3$ (low values above northern Italy, enhanced values over northern Europe) is reproduced by the model, but maximum values at 12:00

UTC are lower in the model (700 ppbv) compared to the measurement (1400 ppbv).

For PGS14 (Fig. 8b,e,h), the measured distribution of $O_3$ over the course of the flight is well reproduced by CLaMS, and also fine structures (e.g. at 18:00 UTC) are clearly visible in both data sets. For the enhanced $O_3$ VMRs in the middle of the flight (15:00-17:00 UTC), the absolute values are measured to be higher but agree within their total estimated errors with CLaMS. In this region, structures are also visible in the model data that are less pronounced in the measurements. At $\theta =$

300-350 K altitude, higher VMRs are visible in the model data compared to the GLORIA observations. The $O_3$ VMR values simulated by EMAC reproduce the overall measured vertical structure, although finer features in the trace gas distributions are not visible due to the horizontal resolution of EMAC ($\approx$125 km) compared to GLORIA (along-track sampling $\approx$3 km, horizontal resolution along viewing direction several $10-100$ km, see also Woiwode et al. (2018)).

During the late winter PGS21 flight (Fig. 8c,f,i), again the modeled and measured two dimensional distributions of $O_3$

generally agree between CLaMS and GLORIA, and again higher VMR values are modeled than observed at $\theta = 300-350$ K





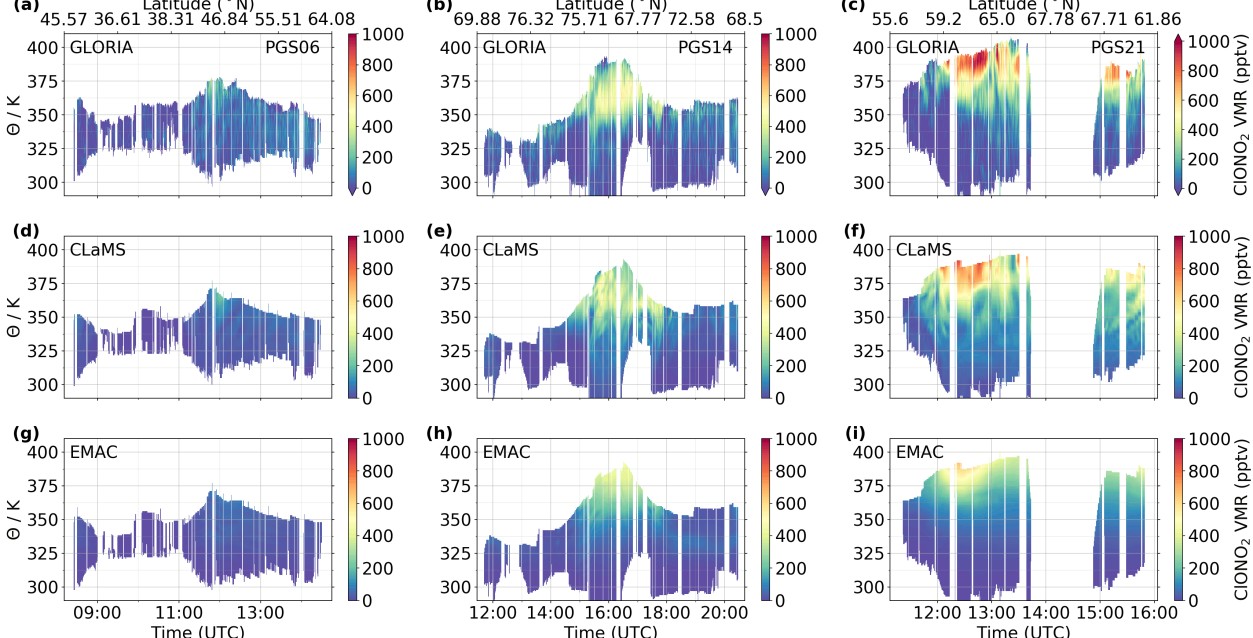

**Figure 9.** Comparison of GLORIA measured (first row, repeated from Figs. 5-7 with a different color bar) to CLaMS (second row) and EMAC (third row) simulated $ClONO_2$ cross sections for flights PGS06 (12 January 2016, first column), PGS14 (26 February 2016, second column) and PGS21 (18 March 2016, third column). Colorbars with top/bottom arrowheads indicate data greater/smaller than the colorbar range.

altitude. Here again finer structures are visible in the CLaMS data, and higher absolute VMRs (>1600 ppbv) are measured than modeled (1100 ppbv) at 400 K, with the difference between them comparable to the total estimated error of the GLORIA data. The EMAC simulation of this flight shows maximum values of $O_3$ (up to 1500 ppbv) that are marginally higher than those from CLaMS and closer to those measured by GLORIA. Again EMAC largely succeeds in reproducing the measured

5 two dimensional trace gas distribution, but with less detail in the small-scale structures than measured by GLORIA (see also Khosrawi et al., 2017).

$ClONO_2$ on flight PGS06 shows the same maximum of 250 pptv at $\theta = 370$ K altitude (12:00 UTC) for CLaMS as for GLORIA (see Fig. 9a,d). Two-dimensional structures with weak $ClONO_2$ enhancements are found at lower altitudes and show slightly different patterns in the two data sets. In EMAC, the simulated enhancement of $ClONO_2$ is barely visible with the

10 color bar used, but is < 100 pptv, smaller than that measured (Fig. 9a,g).

For flight PGS14, again the $ClONO_2$ small scale structures of GLORIA and CLaMS generally agree (see Fig. 9b,e). The maxima found in the $ClONO_2$ distribution around 16:00 UTC coincide in both data sets in terms of the position and absolute values. Compared to the GLORIA measurements, more structures in the area where $ClONO_2$ values are largest in the CLaMS data (as for $O_3$), but all measured structures are also apparent in the model. The comparison to EMAC (see Fig. 9b,h) shows





similar maximum values, but the two dimensional structure of ClONO$_2$ is different: The measured local maximum below flight altitude is not reproduced by the model, which shows the ClONO$_2$ maximum at 370 K flight altitude, while GLORIA shows a local maximum at 360 K.

Flight PGS21 shows higher values in measured ClONO$_2$ compared to CLaMS (see Fig. 9c,f). Besides the overall agreement

of the data, the increased ClONO$_2$ values at $\theta$ = 330-350 K altitude in the model simulation do not match the measured patterns well. For this flight, EMAC simulates the maximum of ClONO$_2$ at a similar position compared to the measurements, and most measured enhancements are also evident in the model data (see Fig. 9c,i). Though this agreement in structure is within expectations given the horizontal resolution of EMAC, the absolute values of ClONO$_2$ in the model (up to 600 pptv) differ substantially from the measured VMRs (1100 pptv), indicating an underestimation of ClONO$_2$ in the model simulation.

These comparisons show that the measured two-dimensional structures in O$_3$ and ClONO$_2$ are well captured in the CLaMS simulation, albeit with some differences in the magnitude of the features. The comparisons to EMAC show general agreement with the measurements, considering the spatial resolution of the model. CLaMS is a Lagrangian CTM, in which transport and chemical reactions are simulated along trajectories following specified meteorological fields. EMAC is an Eulerian CCM, which uses a defined grid on which the chemical processes are simulated. Transport and dynamical processes are taken into

account by the coupling between grid points (Khosrawi et al., 2005; Morgenstern et al., 2017). These different model approaches are also reflected in the comparison of simulated trace gases to GLORIA measurements. While the EMAC simulation used here has been performed on a grid with 1.125° × 1.125° resolution, in the Lagrangian model CLaMS the number of trajectories in the region of interest can be optimized for advanced interpolation methods. For this reason EMAC succeeds in generally reproducing the chemical composition (for O$_3$ and ClONO$_2$) of the measured UTLS regions, while CLaMS is able to

reproduce even small-scale structures in these trace gases. Better agreement is expected for an EMAC simulation with higher horizontal resolution (e.g., T255). Improvements are desirable for both models in the simulation of ClONO$_2$ in the late winter (flight PGS21), when maximum measured VMRs are underestimated. EMAC is known to underestimate downward transport in the lower parts of the polar vortex (Brühl et al., 2007; Khosrawi et al., 2017), and also CLaMS shows diabatic descent that is too weak towards the end of the Arctic winter (see Sec. 3.2). Together with the strong vertical gradient of both gases in

this region, less downward transport of the model results in smaller trace gas abundances at lower altitudes. Differences in maximum values between GLORIA and EMAC for flight PGS06 appear in both discussed trace gases and can most likely be attributed to the limited horizontal resolution of EMAC, because the enhancement seen on flight PGS06 appears to be spatially very confined. CLaMS data for flight PGS21 shows higher VMRs of O$_3$ at $\theta$ = 325 K and of ClONO$_2$ at $\theta$ = 340 K compared to the measurements, despite the observed lack of descent at 380 K. This enhanced CLaMS O$_3$ is also visible in the

passive O$_3$ tracer (not shown), which indicates that this disagreement with the GLORIA measurements is caused by problems in horizontal transport or in the lower boundary conditions of CLaMS. These problems are not unexpected, as CLaMS is a stratospheric model by design. In addition, mixing is difficult to model, but CLaMS has been proven to successfully reproduce mixing during the Arctic winter 2015/16 (Krause et al., 2018).





## 4   CLaMS investigations of chemical evolution

The measurements presented in Sec. 3 revealed unusual chlorine deactivation in the satellite time series and interesting mesoscale structures with unusually high ClONO$_2$ VMRs for the Arctic in the GLORIA measurements. We have shown that CLaMS successfully reproduces structures in the LMS measured by MLS and GLORIA. For this reason, these validated

model simulations are applied to examine the influence of ozone depletion and PSC sedimentation on chlorine deactivation. In the second part of this section, the origin of measured ClONO$_2$ in the LMS is investigated.

### 4.1   Influence of ozone depletion and denitrification on chlorine deactivation

NO$_y$ and O$_3$ abundances are known to have a major influence on chlorine deactivation pathways, and those abundances are strongly affected by denitrification and ozone loss, respectively. In Sec. 3.1, the partitioning of chlorine reservoirs in 2016 was

identified to be unusual for an Arctic winter, and therefore CLaMS sensitivity simulations have been performed to understand and quantify the influence of ozone depletion and PSC sedimentation on chlorine deactivation.

The sensitivity run without ozone depletion has been facilitated by replacing the CLaMS O$_3$ with the passive O$_3$ tracer at the beginning of each simulation step. This passive O$_3$ tracer is initialized, transported and mixed in the same way as the regular O$_3$ field, but it does not experience ozone-depleting processes. The difference between the passive O$_3$ and the standard

O$_3$ is a measure of chemical ozone loss and is presented as time series at 380 K and 490 K (Fig. 10a) and as a cross section between 330 K and 600 K (Fig. 10c). These difference time series show that ozone depletion starts in the beginning of January and reaches its maximum in the middle of March. The largest ozone depletion is simulated at 490 K, with maximum VMR differences of 1.75 ppmv. The influence of ozone depletion on HCl and ClONO$_2$ is illustrated in Fig. 10e,g as differences between the reference and the sensitivity simulation without ozone depletion. Negative differences (shades of blue) indicate

how much the chlorine reservoir is diminished due to the effect of ozone depletion, while positive differences (shades of red) show enhancements due to this effect. HCl exhibits a positive response to ozone depletion, and starting from the beginning of March, more than 500 pptv additional chlorine is deactivated into HCl under ozone-depleted conditions. At the same time, ClONO$_2$ is reduced by more than 500 pptv due to ozone depletion. These differences for both reservoir gases peak at altitudes of 440 K towards the end of March. Interestingly, this peak altitude is lower than the altitude of greatest ozone loss (490 K).

For the sensitivity run without PSC sedimentation, a CLaMS simulation was performed without the sedimentation module (see Sec. 2.3.1). Differences in NO$_y$ between the reference and the sensitivity simulation are presented in Fig. 10b,d as time series at 380 K and 490 K and as cross sections. Nitrification up to 4 ppbv is seen at 380 K in early winter and later at lower altitudes, while denitrification is largest (up to 10 ppbv) at 490 K in the middle of January. The effect on HCl and ClONO$_2$ (Fig. 10f,h) shows an enhancement of HCl, which reaches its maximum (500 pptv) in the middle of February, and a decrease

in ClONO$_2$, which reaches its extreme (> 500 pptv) in the beginning of March. A weak opposite effect is observed at lower altitudes (< 380 K), where nitrification is observed.

Absolute values of HCl, ClONO$_2$, and ClO$_x$ at 380 K and 490 K are presented in Fig. 10i,j for the reference and both sensitivity simulations. At 380 K, differences up to 200 pptv between the reference and the O$_3$ sensitivity simulation ("+")





**Figure 10.** CLaMS vortex average (equivalent latitude > 75°N) time series for ozone loss at 380 K and 490 K (a), and as cross section (c). Denitrification is presented in the same manner (b,d). Differences in HCl and ClONO₂ between a reference simulation and a simulation without the influence of ozone depletion (e,g) or PSC sedimentation (f,h). Altitudes of 380 K and 490 K, which are illustrated in other panels, are marked with horizontal lines. The bottom panels show ClO$_x$ (black), HCl (green), and ClONO₂ (orange) for the reference simulation (open squares), and for the sensitivity simulations without ozone depletion (+) and without PSC sedimentation (x) at 380 K (i) and 490 K (j). Please note that these time series, in contrast to others in this figure, start in January.





are visible in the reservoirs in March, with more $ClONO_2$ and less HCl in the simulation without ozone depletion. PSC sedimentation ("x") has only a weak influence at 380 K on the temporal evolution of the chlorine reservoirs and $ClO_x$, with small differences visible around the time of the final warming in the beginning of March. At 490 K, differences in the reservoirs between the reference and the $O_3$ sensitivity run increase to 300 pptv, starting in the beginning of March. Again, more $ClONO_2$

and less HCl is modeled for the sensitivity simulation without ozone depletion. A larger change in the chlorine partitioning is observed for the sensitivity simulation without PSC sedimentation: Starting towards the end of January, less HCl and more $ClO_x$ is simulated, and by the beginning of March a substantial increase in $ClONO_2$ with a difference of 1000 pptv compared to the reference is found. During that time, $ClO_x$ decreases faster and HCl is consistently lower in comparison to the reference simulation. At the end of March, the sensitivity simulation without PSC sedimentation approaches the reference simulation for

all presented species.

The sensitivity simulations by CLaMS help to quantify the effect of ozone depletion and PSC sedimentation on the observed unusual chlorine deactivation in 2016. The sensitivity simulation without ozone depletion showed that at 380 K, low ozone abundances (< 1.0 ppmv) caused 200 pptv of chlorine to be deactivated into HCl instead of $ClONO_2$. These ozone abundances are not as low (< 0.5 ppmv) as those found in previous studies (Prather and Jaffe, 1990; Douglass et al., 1995; Grooß et al.,

1997, 2005; Mickley et al., 1997), but as demonstrated by Douglass and Kawa (1999), even higher ozone abundances than 0.5 ppmv together with cold temperatures are able to change chlorine deactivation. The different altitudes of the maxima observed in ozone loss (490 K) and changes in chlorine species (440 K) may be explained by the vertically increasing ozone VMR profile in the LMS. At 490 K, the absolute chemical ozone loss (≈1.75 ppmv) is larger than at 440 K (≈1.2 ppmv), but the total ozone VMR still is larger at 490 K (≈2.5 ppmv) than at 440 K (≈1.5 ppmv, see also Fig. 3). According to Douglass

and Kawa (1999), the absolute ozone VMR is important for the chlorine deactivation partitioning, and not the chemical ozone loss. The availability of $NO_y$ does not play a significant role in chlorine partitioning at 380 K. This may be explained by the fact that CLaMS does not simulate denitrification at 380 K (see Fig. 10b), and the availability of $NO_y$ is not limited due to PSC sedimentation at this level. Thus, there is little difference between the control and sensitivity simulations in this case. At 490 K, the reduced availability of $O_3$ caused a relatively small proportion of chlorine (300 pptv) to be deactivated into HCl

instead of $ClONO_2$. The decreased availability of $NO_y$ (as a consequence of PSC sedimentation) induces major differences up to 1000 pptv in $ClONO_2$.

The two sensitivity simulations reveal different time and altitude ranges in which they have the strongest impact on chlorine deactivation: While $O_3$ abundances affect the partitioning of the reservoirs from the beginning of March (with a maximum towards the end of March), the availability of $NO_y$ has the maximum effect on HCl between February and the middle of March

and on $ClONO_2$ around the major warming in the beginning of March. These sensitivity simulations also show impacts over different altitude ranges: The effect of $O_3$ depletion leads to notable differences between reference and sensitivity simulation starting from 380 K, with maximal differences at 440 K. The effect of sedimentation of $HNO_3$-containing particles on the chlorine reservoirs shows large impact between 400 K and 550 K. A small opposite effect is observed at altitudes below 380 K due to re-nitrification. The differences in HCl and $ClONO_2$ for ozone sensitivity appear to be very symmetric: positive

differences in HCl correspond to negative differences in $ClONO_2$ at approximately the same time and altitude and with roughly

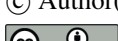


the same magnitude. This is because $O_3$ abundances influence the partitioning of Cl and ClO, which directly determines whether chlorine is deactivated into HCl or $ClONO_2$ (Douglass and Kawa, 1999). For the sensitivity to PSC sedimentation, this symmetry between the differences in the chlorine reservoirs is not observed. Without PSC sedimentation, the chlorine activation is already changed by the end of January, while PSCs are present and chlorine activation is still possible. During

this time, in the absence of denitrification, more $ClONO_2$ is produced due to the greater availability of $NO_y$, assuming that there is sufficient sunlight in the vortex to photolyze $HNO_3$ to produce $NO_2$. Together with available HCl, this regenerated $ClONO_2$ is then activated on PSCs, which results in net chlorine activation. In March, chlorine is deactivated into $ClONO_2$ to a considerably larger extent, again due to the greater availability of $NO_y$. The impact over different altitude ranges of the two sensitivity simulations implies that the observed unusually strong chlorine deactivation into HCl at 380 K (see Sec. 3.1) was

predominantly driven by low $O_3$. At 490 K, where denitrification was much stronger than at 380 K, it was mainly the low $NO_y$ that shifted chlorine deactivation towards HCl, while the low $O_3$ abundances only played a minor role.

The passive $O_3$ tracer of CLaMS also allows estimation of chemical ozone loss of 0.4 ppmv at 380 K and 1.75 ppmv at 490 K. Since the comparisons to MLS indicate that CLaMS overestimates $O_3$ towards the end of the winter, possibly because of deficiencies in its representation of dynamical processes, these estimates of ozone loss should be regarded as a lower

boundary. Comparisons based on MLS data show that only a few other Arctic winters have experienced chemical ozone loss in the LMS as large as that in 2015/16 (Livesey et al., 2015; Santee, 2017).

## 4.2   Origin of $ClONO_2$ measured by GLORIA

In order to investigate the temporal evolution of the chemical composition at geolocations measured by GLORIA, CLaMS is used to calculate 11-day backward trajectories from these measurement geolocations. Then the model variables from the

global model run are interpolated to these geolocations 11 days before the measurement, and CLaMS performs its Lagrangian simulation along the trajectory leading to the measurement. Along this trajectory, variables are saved at a temporal resolution of one hour. Because the chemical composition is simulated only along the trajectories, mixing was not considered for this simulation. As discussed by Konopka et al. (2003), mixing is regarded to have a weak influence on chlorine deactivation. This explains differences in CLaMS $ClONO_2$ cross sections between Figs. 9 and 11. A similar approach to investigate chlorine

activation along backward trajectories has been reported by Lelieveld et al. (1999), but based on in-situ measurements of HCl and with a focus on chlorine activation on cirrus clouds.

The cross sections at the trajectory ending points are shown in Fig. 11(a-d) for flight PGS14 for $ClONO_2$ and HCl as the reservoir gases and for ClO as one of the major active chlorine species at these altitudes. According to the validated $ClONO_2$ cross section, regions of interest are identified and marked: The local maximum of $ClONO_2$ at 16:00 UTC and $\theta = 360$ K

is marked with a magenta "star" symbol, another local maximum at 16:40 UTC and $\theta = 370$ K is marked green and the last substantial local maximum at 17:45 UTC and $\theta = 355$ K is marked blue. In the ClO cross section a maximum is modeled at 16:35 UTC and $\theta = 385$ K which is marked cyan. CLaMS backward trajectories within a horizontal distance of 25 km and a vertical distance of $\theta = 5$ K from these marked geolocations are selected for further analysis. These selected trajectories are





projected on a map in Fig. 11d. It can be seen that the majority of these air parcels stay confined within a well defined region which is expected to be the polar vortex.

For flight PGS21, results from this trajectory analysis are presented in the same manner (Fig. 11e-h). Regions of interest are identified at different locations of enhanced $ClONO_2$ values at 12:10 UTC and $\theta = 380$ K (red), 13:25 UTC and $\theta = 390$ K (dark blue) and 15:20 UTC and $\theta = 380$ K (light green). Trajectories in the vicinity of these points are selected as described for flight PGS14. In the map projection of these trajectories (Fig. 11h), it can be seen that these air masses have been confined above Greenland (red) or have been circulating above Siberia (dark blue, light green) until they migrated to Scandinavia, where they were measured. The tracks of these trajectories are consistent with the meteorological situation of an eroding polar vortex during the time of flight PGS21.

The temporal evolution along these selected trajectories is shown for both flights in Fig. 12 for (a) potential temperature, (b) the solar zenith angle (SZA), (c) $ClONO_2$, (d) HCl, (e) $ClO_x$ (= $ClO+2Cl_2O_2+2Cl_2$), (f) ClO, (g) $Cl_2O_2$ and (h) $Cl_2$. The mean of all selected trajectories belonging to a point of interest is presented as a solid line in the corresponding color, while minimum and maximum values are marked with shading in the same color.

### 4.2.1 Flight on 26 February 2016 (PGS14)

Potential temperatures (Fig. 12a1) show persistent downwelling along all selected trajectories, and the SZA (Fig. 12b1) indicates long periods in darkness. These first two panels provide context for the temporal evolution of the chlorine species.

The magenta curves, which have been defined to end at the region with the highest $ClONO_2$ in the cross section, show persistently high VMRs for $ClONO_2$ between 250 and 500 pptv and HCl values around 375 pptv. These persistently high VMRs in the chlorine reservoir species indicate that chlorine deactivation mainly occurred prior to the end point of the 11-day back trajectory and that these deactivated air masses have been transported to the GLORIA measurement location. Both chlorine reservoirs slightly increase during the first four days until 18 February 2016, then decrease for three days (until 20 February 2016) and then increase again, with a stronger increase seen in $ClONO_2$. The active chlorine species (as a sum shown in $ClO_x$) for the magenta curves remain at low levels (< 250 pptv), and show enhancements when the chlorine reservoirs are at lower values.

The green curves end at a local maximum of 340 pptv of $ClONO_2$ at $\theta = 370$ K. This enhanced value was reached due to an increase within the last five days. In the beginning of these 11-day trajectories, $ClONO_2$ started at VMRs of approximately 375 pptv until 17 February 2016 and then decreased to minimum values on 20 February 2016. The temporal evolution of HCl shows a similar slope: The mean value of these trajectories starts at a VMR of 470 pptv, which increases on the same time scale as $ClONO_2$ to 500 pptv, decreases to 125 pptv and finally increases to 320 pptv. During the time of $ClONO_2$ and HCl decrease, $Cl_2$ builds up to 270 pptv until it suddenly decreases to 0 pptv on 21 February 2016. At the same time, which coincides with the first exposure of this air parcel to sunlight since 16 February 2016 (according to the SZA), the VMR of ClO rapidly increases and then decreases again (with signatures of its diurnal cycle), while the reservoirs ($ClONO_2$ and HCl) increase again along with the decrease of $ClO_x$. The ClO-dimer increases first on 17 February 2016, when $Cl_2$ starts to rise, and increases again on



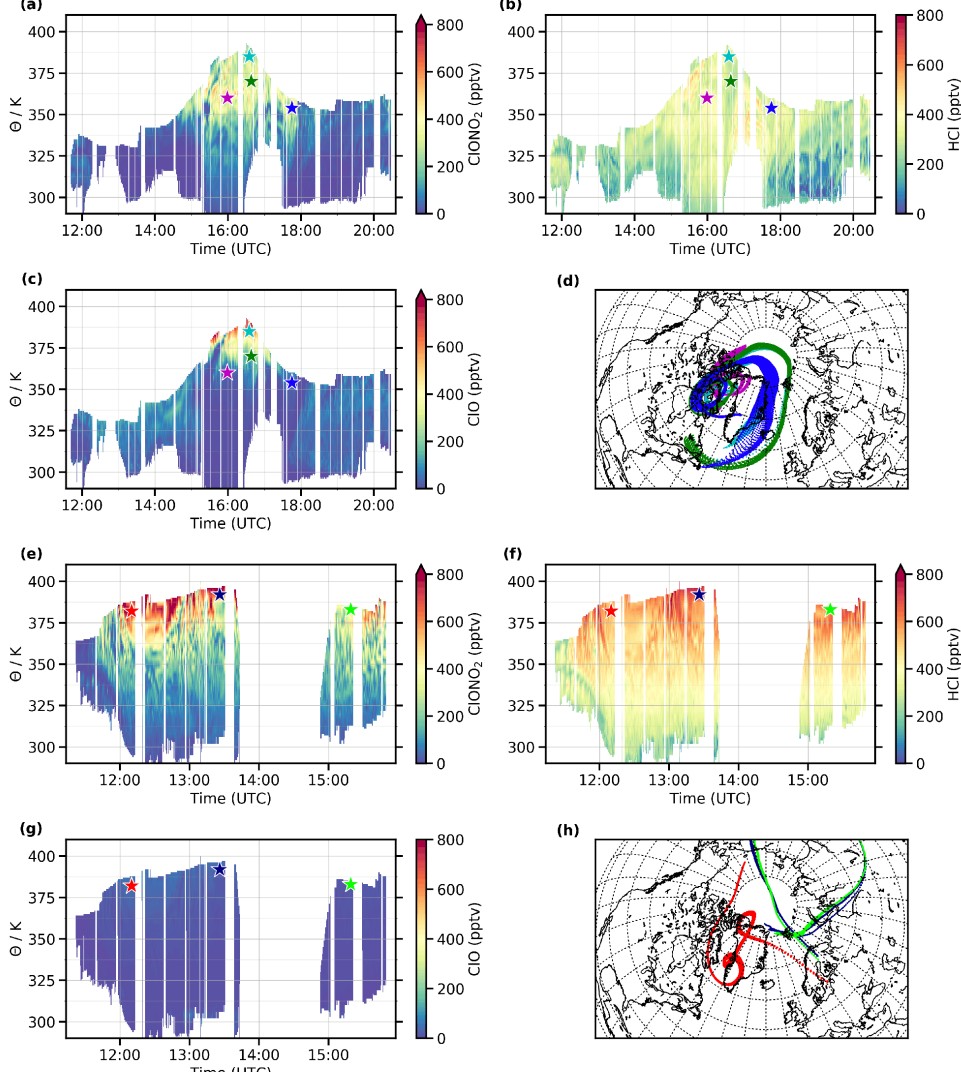

**Figure 11.** Cross sections for (a,e) $ClONO_2$, (b,f) HCl and (c,g) ClO at the end of CLaMS trajectories leading to the GLORIA tangent point geolocations for flights PGS14 (a-d) and PGS21 (e-h). Regions of interest are marked with colored star symbols. 11-day backward trajectories calculated for each flight by CLaMS are shown on the map (d,h) in corresponding colors.

18 February 2016. In addition to the substantial increase of ClO on 21 February 2016, also the ClO-dimer rapidly increases to 100 pptv but then decreases again.

The blue curves show a similar course compared to the green ones but at lower potential temperature altitudes. Due to these different altitudes, the blue curves start with lower availability of chlorine reservoirs. At first (17 February 2016 and beginning of 18 February 2016) $Cl_2$ along the blue trajectory exceeds that on the green one, but ultimately less chlorine is activated (see





**Figure 12.** Modeled temporal evolution of (a) potential temperature, (b) solar zenith angle (SZA, the 90° threshold is marked with a black line), (c) $ClONO_2$, (d) $HCl$, (e) $ClO_x$, (f) $ClO$, (g) $Cl_2O_2$, and (h) $Cl_2$ for (1) flight PGS14 (left) and (2) flight PGS21 (right). These trajectories are color-coded as defined in Fig. 11 a-c and e-g. Solid lines show the mean of all trajectories connected with these regions and the light colors show minimum/maximum values among those trajectories. Please note different ordinate scales in each column.

$Cl_2$), because the blue parcels saw less sunlight than the green ones on 16 February 2016 and none on 17 February 2016. This also results in less $ClO$ after the exposure to sunlight (21 February 2016), but these lower amounts of $ClO$ are initially



deactivated slightly more rapidly into ClONO$_2$ compared to the green curve. During the last three days of this trajectory, the green and the blue curves are almost identical for the chlorine reservoirs.

As an example of enhanced ClO values at the end of the trajectories, the cyan curves (which are at higher potential temperature altitudes of 385 K) show also a similar course compared to the green curves during the last five days before the

measurement. In the beginning of the presented 11 days, the reservoir gases decrease earlier (16 February 2016) to low VMR levels than they did for the green point. ClO$_x$ values are highest compared to other trajectory sets. The amount of accumulated Cl$_2$ is similar to that in the blue parcels (175 pptv) with the same times of sunlight exposure on 21 February 2016 and 22 February 2016. ClO also shows a similar evolution compared to the green course (if one considers the different altitudes, which are presumably the reason for the higher VMRs in the beginning) until the last three days, when ClO increases considerably along

the cyan trajectory. This increase goes along with a decrease of the ClO-dimer, which is significantly higher along the entire trajectory compared to all other curves, reaching mean values up to 440 pptv on 20 February 2016.

For all selected points, there are regions in which the minimum/maximum values (indicated by the colored shading) vary substantially from the mean value, which indicates strong variability even for trajectories that end at geolocations within a horizontal distance of 25 km and $\theta$ = 5 K altitude.

### 4.2.2   Flight on 18 March 2016 (PGS21)

In contrast to flight PGS14, sunlight is available along all selected trajectories of flight PGS21 on a daily basis, according to the SZA (Fig. 12b2). Based on the potential temperature, continuous subsidence is observed for the red trajectory set, while for the light-green and dark-blue trajectories the air parcels are slightly uplifted until 13 March 2016 and then subside with a similar slope compared to the other trajectories. ClO$_x$ (Fig. 12e2) predominantly consists of ClO for all selected trajectories

during that time of the year.

The red set of trajectories was selected due to the enhanced ClONO$_2$ values at the measurement location. The history of this enhancement in ClONO$_2$ shows a slight overall decrease (from 760 pptv to 630 pptv) with decreasing and increasing features due to the diurnal cycle. HCl increases step-wise from 380 pptv to 500 pptv, and ClO$_x$ shows diurnal enhancements during the sunlit periods up to 60 pptv.

For the light-green air parcels, ClONO$_2$ slowly decreases from 900 to 750 pptv with fluctuations due to the diurnal cycle. HCl starts at 400 pptv and ends at 600 pptv by increasing in small steps. ClO shows a small diurnal cycle with maximum values up to 100 pptv.

The highest potential temperature trajectory end point for this flight is marked in dark blue. ClONO$_2$ increases from 670 to 870 pptv and then decreases again to 700 pptv with fluctuations due to the diurnal cycle. These dark-blue trajectories show the

largest increase in HCl (from 450 pptv to 700 pptv). The diurnal cycle also dominates the evolution of ClO, with maximum values up to 200 pptv superimposed on a baseline value of 100 pptv that persists until 13 March 2016. During this period also small (50 pptv) remnants of Cl$_2$O$_2$ are visible.

For most of these selected trajectory sets it can be observed that the variability is smaller compared to the ones for PGS14, especially for chlorine species other than ClONO$_2$.




### 4.2.3 Discussion

According to chemical tracers along CLaMS backward trajectories, enhanced $ClONO_2$ measured by GLORIA on 26 February 2016 was mainly a result of chlorine deactivation within the last five days before the measurement. We also presented an example of air masses that had been deactivated prior to the end point of the 11-day back trajectory and then transported to the

GLORIA measurement location. Our results also revealed substantial variability among trajectories initialized within a given region of interest (marked by the shaded area: 25 km horizontal and $\theta = 5$ K vertical coincidence). This variability indicates that small changes in the ending point of the trajectories result from different chemical histories of the air parcels.

For flight PGS21 in mid March, the $ClONO_2$ along the trajectories shows constantly high VMRs, modulated by the diurnal cycle. Since this flight took place well after the final warming (5-6 March; Manney and Lawrence, 2016), there had been no

recent PSC formation or chlorine activation, and almost all of the measured enhanced $ClONO_2$ had been produced by chlorine deactivation that took place at least 11 days before the measurement. Changes in $ClONO_2$, $ClO_x$ (which is nearly all ClO at this time) and HCl can be explained by the photolysis of $ClONO_2$, which results in a diurnal cycle (Brasseur and Solomon, 2005). This photolysis diminishes $ClONO_2$ and creates Cl (or a small fraction of ClO) during sunlit portions of the trajectories (Burkholder et al., 2015). These products can either react in ozone loss cycles, during which the fractions of Cl and ClO

may change (see Solomon, 1999), or build chlorine reservoirs again: ClO reacts with $NO_2$ to $ClONO_2$, while Cl reacts to HCl. Therefore not all of the $ClONO_2$ photolysis products ultimately go on to regenerate $ClONO_2$. Thus, HCl increases in a stepwise fashion, while $ClONO_2$ decreases. As was demonstrated with CLaMS sensitivity simulations, preferential deactivation into HCl is caused by low $O_3$ abundances as a consequence of ozone depletion.

In summary, for the most part enhanced $ClONO_2$ measured in February had recently been (within the prior few days)

deactivated in-situ in the LMS, while in March almost all of the measured enhanced $ClONO_2$ had been transported longer than 11 days.

## 5 Conclusions

This study analyzes chlorine activation and deactivation in the Arctic winter 2015/16 LMS by utilizing time series of satellite measurements, aircraft remote sensing measurements from GLORIA during the PGS campaigns and simulations by the atmo-

spheric models CLaMS and EMAC. The analysis of ACE-FTS and MLS time series shows the extreme nature of the Arctic winter 2015/16: The time series of HCl has defined a new minimum in the Arctic at $\theta = 380$ K during the Aura/MLS epoch, followed by an unusually rapid increase in HCl. For several parts of the winter, ClO also showed maximum values within the instrument's record. Although ozone loss was greater in 2010/11, 2015/16 was a winter with extraordinary large chemical ozone loss (Livesey et al., 2015; Santee, 2017), estimated by CLaMS to be at least 0.4 ppmv at 380 K and 1.75 ppmv at 490 K.

The (for the Arctic) unusual chlorine deactivation has been identified through CLaMS sensitivity studies to result at 380 K from low $O_3$ abundances rather than from low $NO_y$ availability caused by PSC sedimentation. At higher potential temperatures (as shown at 490 K), denitrification played a greater role.





During this exceptional Arctic winter, the GLORIA instrument observed strongly enhanced $ClONO_2$ up to 1100 pptv in the LMS. GLORIA measurements of both $O_3$ and $ClONO_2$ show mesoscale structures in the two dimensional vertical cross sections. The comparisons of the highly resolved GLORIA cross sections of $O_3$ and $ClONO_2$ with the models EMAC and CLaMS are useful evaluations of two different approaches (Eulerian CCM and Lagrangian CTM) for modeling the chemical
composition of the UTLS. The comparison of EMAC model data with measurements shows agreement within the limitations expected due to its relatively coarse resolution compared to CLaMS and GLORIA. In addition, well-known problems of EMAC's diabatic descent are observed in the comparisons. CLaMS benefits from its higher spatial resolution and reproduces the measurements even for detailed small-scale structures. For the late winter flight PGS21, CLaMS shows more $O_3$ and $ClONO_2$ than GLORIA at lower potential temperature altitudes (330-340 K), which indicates that CLaMS could be improved
concerning boundary conditions at low altitudes, horizontal transport and mixing. Discrepancies at 380 K between measurements (MLS $CH_3Cl$ and GLORIA $O_3$ and $ClONO_2$) and CLaMS data also suggest potential for improvement in the model's representation of diabatic descent in the LMS. Generally, comparisons of CLaMS with MLS demonstrate overall agreement, while differences in HCl reflect well known problems with modeled chlorine activation, which also have consequences for $O_3$ and ClO.

The origin of observed enhanced $ClONO_2$ at selected points in the LMS is reconstructed with an analysis of the chemical composition along trajectories leading to the measurement geolocations provided by CLaMS. As expected, this analysis shows that both transport of $ClONO_2$ and in-situ deactivation at lower altitudes are simulated in the model for the selected February flight. Large variability among trajectories ending in the same vicinity shows that small changes in the path of an air parcel may strongly influence its course of chlorine activation/deactivation due to different encounters with PSCs, and different amounts
of available $NO_y$, $O_3$, and sunlight. For the flight in March 2016, the enhanced measured $ClONO_2$ is a result of transport, as illustrated by the chemical composition along the CLaMS trajectories.

Due to climate change, exceptionally cold winters are expected to occur more frequently in the future (Fels et al., 1980; Hartmann et al., 2014; WMO, 2015), which may in particular impact ozone in the Arctic LMS. These expected changes in ozone and chlorine activation and deactivation processes emphasize the importance of regular observations of the chemical
composition of the atmosphere, with a particular focus on the LMS.

*Data availability.* The discussed GLORIA data set is available at the HALO database (https://doi.org/10.17616/R39Q0T) and at the KITopen repository (https://doi.org/10.5445/IR/1000086506). Aura/MLS and MERRA2 data are available at the Goddard Earth Sciences Data and Information Services Center (https://doi.org/10.5067/AURA/MLS/DATA2017, https://doi.org/10.5067/AURA/MLS/DATA2012, https://doi.org/10.5067/AURA/MLS/DATA2010, https://doi.org/10.5067/AURA/MLS/DATA2004, https://doi.org/10.5067/AURA/MLS/DATA2002, and
https://doi.org/10.5067/A7S6XP56VZWS. ACE-FTS data is located at https://database.scisat.ca/level2/, registration required. Data from the EMAC and CLaMS simulations are available from the authors upon request.





*Author contributions.* SJ initiated the study, performed the analyses and wrote the manuscript. MLS aided with the handling and interpretation of Aura/MLS data, significantly contributed to the analyses, and refined the manuscript. JUG and IT performed the CLaMS simulations and helped to interpret this model data. MH, JU, WW, MB and SJ processed and analyzed GLORIA data. FFV and EK together with the GLORIA team performed the GLORIA measurements during the PGS campaigns. FK and OK performed and interpreted the EMAC simulations. KAW helped with the usage and interpretation of ACE-FTS data. HO, BMS and WW organized and coordinated the POLSTRACC campaign. All authors assisted with the interpretation of the results and writing of the manuscript

*Competing interests.* The authors declare that they have no conflict of interest.

*Acknowledgements.* We gratefully thank the PGS coordination team and the DLR-FX for successfully conducting the field campaign. The results are based on the efforts of all members of the GLORIA team, including the technology institutes ZEA-1 and ZEA-2 at Forschungszentrum Jülich and the Institute for Data Processing and Electronics at the Karlsruhe Institute of Technology. We thank Michael C. Pitts for providing CALIPSO/CALIOP data for the 2015/16 cloud area time series. The authors gratefully acknowledge the computing time for the CLaMS simulations granted on the supercomputer JURECA at Jülich Supercomputing Centre (JSC) under the VSR project ID JICG11. We thank NASA for providing their MERRA2 meteorological reanalysis data set. S. Johansson has received funding from the European Community's Seventh Framework Programme (FP7/2007-2013) under grant agreement 603557. S. Johansson gratefully thanks the Graduate School for Climate and Environment (GRACE), Karlsruhe Institute of Technology for funding his visit to the Jet Propulsion Laboratory to discuss the Aura/MLS measurements, and the MLS team for the hospitality during that time. Work at the Jet Propulsion Laboratory, California Institute of Technology, was done under contract with the National Aeronautics and Space Administration. The Atmospheric Chemistry Experiment (ACE), also known as SCISAT, is a Canadian-led mission mainly supported by the Canadian Space Agency. This work was partly supported by the German Research Foundation (Deutsche Forschungsgemeinschaft, DFG Priority Program SPP 1294). We acknowledge support by the Deutsche Forschungsgemeinschaft and the Open Access Publishing Fund of the Karlsruhe Institute of Technology.





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

©c Author(s) 2019. CC BY 4.0 License.





Roeckner, E., Brokopf, R., Esch, M., Giorgetta, M., Hagemann, S., Kornblueh, L., Manzini, E., Schlese, U., and Schulzweida, U.: Sensitivity of Simulated Climate to Horizontal and Vertical Resolution in the ECHAM5 Atmosphere Model, J. Climate, 19, 3771–3791. https://doi.org/10.1175/JCLI3824.1, 2006.

Sander, S. P., Friedl, R. R., Barker, J. R., Golden, D. M., Kurylo, M. J., Wine, P. H., Abbatt, J., Burkholder, J. B., Kolb, C. E., Moortgat, G. K., Huie, R. E., and Orkin, V. L.: Chemical kinetics and photochemical data for use in atmospheric studies, Evaluation no. 17, JPL Publication, 10-6. http://jpldataeval.jpl.nasa.gov, 2011.

Santee, M. L.: Quantifying and confirming our understanding of processes controlling stratospheric ozone: Insights from UARS and Aura MLS. http://www.montreal30.io3c.org/sites/montreal30.io3c.org/files/pictures/19apm/Santee_MPSymp-MLS_Overview.pdf, 2017.

Santee, M. L., Tabazadeh, A., Manney, G. L., Salawitch, R. J., Froidevaux, L., Read, W. G., and Waters, J. W.: UARS Microwave Limb Sounder HNO3 observations: Implications for Antarctic polar stratospheric clouds, J. Geophys. Res., 103, 13 285–13 313. https://doi.org/10.1029/98JD00365, 1998.

Santee, M. L., Manney, G. L., Livesey, N. J., and Waters, J. W.: UARS Microwave Limb Sounder observations of denitrification and ozone loss in the 2000 Arctic late winter, Geophys. Res. Let., 27, 3213–3216. https://doi.org/10.1029/2000GL011738, 2000.

Santee, M. L., Lambert, A., Read, W. G., Livesey, N. J., Cofield, R. E., Cuddy, D. T., Daffer, W. H., Drouin, B. J., Froidevaux, L., Fuller, R. A., Jarnot, R. F., Knosp, B. W., Manney, G. L., Perun, V. S., Snyder, W. V., Stek, P. C., Thurstans, R. P., Wagner, P. A., Waters, J. W., Muscari, G., de Zafra, R. L., Dibb, J. E., Fahey, D. W., Popp, P. J., Marcy, T. P., Jucks, K. W., Toon, G. C., Stachnik, R. A., Bernath, P. F., Boone, C. D., Walker, K. A., Urban, J., and Murtagh, D.: Validation of the Aura Microwave Limb Sounder HNO3 measurements, J. Geophys. Res. Atmos., 112, 2007JD008 721. https://doi.org/10.1029/2007JD008721, 2007.

Santee, M. L., Lambert, A., Read, W. G., Livesey, N. J., Manney, G. L., Cofield, R. E., Cuddy, D. T., Daffer, W. H., Drouin, B. J., Froidevaux, L., Fuller, R. A., Jarnot, R. F., Knosp, B. W., Perun, V. S., Snyder, W. V., Stek, P. C., Thurstans, R. P., Wagner, P. A., Waters, J. W., Connor, B., Urban, J., Murtagh, D., Ricaud, P., Barret, B., Kleinböhl, A., Kuttippurath, J., Küllmann, H., von Hobe, M., Toon, G. C., and Stachnik, R. A.: Validation of the Aura Microwave Limb Sounder ClO measurements, J. Geophys. Res. Atmos., 113. https://doi.org/10.1029/2007JD008762, 2008a.

Santee, M. L., MacKenzie, I. A., Manney, G. L., Chipperfield, M. P., Bernath, P. F., Walker, K. A., Boone, C. D., Froidevaux, L., Livesey, N. J., and Waters, J. W.: A study of stratospheric chlorine partitioning based on new satellite measurements and modeling, J. Geophys. Res. Atmos., 113. https://doi.org/10.1029/2007JD009057, 2008b.

Santee, M. L., Manney, G. L., Livesey, N. J., Froidevaux, L., Schwartz, M. J., and Read, W. G.: Trace gas evolution in the lowermost stratosphere from Aura Microwave Limb Sounder measurements, J. Geophys. Res. Atmos. 116. https://doi.org/10.1029/2011JD015590, 2011.

Santee, M. L., Livesey, N. J., Manney, G. L., Lambert, A., and Read, W. G.: Methyl chloride from the Aura Microwave Limb Sounder: First global climatology and assessment of variability in the upper troposphere and stratosphere, J. Geophys. Res. Atmos., 118, 13,532–13,560. https://doi.org/10.1002/2013JD020235, 2013.

Sheese, P. E., Walker, K. A., Boone, C. D., McLinden, C. A., Bernath, P. F., Bourassa, A. E., Burrows, J. P., Degenstein, D. A., Funke, B., Fussen, D., Manney, G. L., McElroy, C. T., Murtagh, D., Randall, C. E., Raspollini, P., Rozanov, A., Russell III, J. M., Suzuki, M., Shiotani, M., Urban, J., Clarmann, T. V., and Zawodny, J. M.: Validation of ACE-FTS version 3.5 NOy species profiles using correlative satellite measurements, Atmos. Meas. Tech., 9, 5781–5810. https://doi.org/10.5194/amt-9-5781-2016, 2016.

Solomon, S.: Stratospheric ozone depletion: A review of concepts and history, Rev. Geophys., 37, 275–316. https://doi.org/10.1029/1999RG900008, 1999.





Stiller, G. P., ed.: The Karlsruhe Optimized and Precise Radiative transfer Algorithm (KOPRA), vol. FZKA 6487 of *Wissenschaftliche Berichte*, Forschungszentrum Karlsruhe, 2000.

Tikhonov, A. N. and Arsenin, V. I.: Solutions of ill-posed problems, Scripta series in mathematics, Winston and Distributed solely by Halsted Press, Washington and New York. ISBN: 9780470991244, 1977.

Tritscher, I., Grooß, J.-U., Spang, R., Pitts, M. C., Poole, L. R., Müller, R., and Riese, M.: Lagrangian simulation of ice particles and resulting dehydration in the polar winter stratosphere, Atmos Chem. Phys. Discussions, pp. 1–32. https://doi.org/10.5194/acp-2018-337, 2018.

Ungermann, J., Blank, J., Dick, M., Ebersoldt, A., Friedl-Vallon, F., Giez, A., Guggenmoser, T., Höpfner, M., Jurkat, T., Kaufmann, M., Kaufmann, S., Kleinert, A., Krämer, M., Latzko, T., Oelhaf, H., Olchewski, F., Preusse, P., Rolf, C., Schillings, J., Suminska-Ebersoldt, O., Tan, V., Thomas, N., Voigt, C., Zahn, A., Zöger, M., and Riese, M.: Level 2 processing for the imaging Fourier transform spectrometer
GLORIA: Derivation and validation of temperature and trace gas volume mixing ratios from calibrated dynamics mode spectra, Atmos. Meas. Tech., 8, 2473–2489. https://doi.org/10.5194/amt-8-2473-2015, 2015.

Urban, J., Lautié, N., Le Flochmoën, E., Jiménez, C., Eriksson, P., La Noë, J. d., Dupuy, E., Ekström, M., Amraoui, L. E., Frisk, U., Murtagh, D., Olberg, M., and Ricaud, P.: Odin/SMR limb observations of stratospheric trace gases: Level 2 processing of ClO, N2O, HNO3, and O3, J. Geophys. Res. Atmos., 110. https://doi.org/10.1029/2004JD005741, 2005.

Voigt, C., Dörnbrack, A., Wirth, M., Groß, S. M., Pitts, M. C., Poole, L. R., Baumann, R., Ehard, B., Sinnhuber, B.-M., Woiwode, W., and Oelhaf, H.: Widespread polar stratospheric ice clouds in the 2015–2016 Arctic winter – implications for ice nucleation, Atmos. Chem. Phys., 18, 15 623–15 641. https://doi.org/10.5194/acp-18-15623-2018, 2018.

von Clarmann, T.: Chlorine in the Stratosphere, Atmósfera, 26. http://www.revistascca.unam.mx/atm/index.php/atm/article/download/38656/36823, 2013.

von Clarmann, T., Linden, A., Oelhaf, H., Fischer, H., Friedl-Vallon, F., Piesch, C., Seefeldner, M., Völker, W., Bauer, R., Engel, A., and Schmidt, U.: Determination of the stratospheric organic chlorine budget in the spring arctic vortex from MIPAS B limb emission spectra and air sampling experiments, J. Geophys. Res. Atmos., 100, 13 979–13 997. https://doi.org/10.1029/95JD01048, 1995.

von Hobe, M., Bekki, S., Borrmann, S., Cairo, F., D'Amato, F., Di Donfrancesco, G., Dörnbrack, A., Ebersoldt, A., Ebert, M., Emde, C., Engel, I., Ern, M., Frey, W., Genco, S., Griessbach, S., Grooß, J.-U., Gulde, T., Günther, G., Hösen, E., Hoffmann, L., Homonnai, V.,
Hoyle, C. R., Isaksen, I. S. A., Jackson, D. R., Jánosi, I. M., Jones, R. L., Kandler, K., Kalicinsky, C., Keil, A., Khaykin, S. M., Khosrawi, F., Kivi, R., Kuttippurath, J., Laube, J. C., Lefèvre, F., Lehmann, R., Ludmann, S., Luo, B. P., Marchand, M., Meyer, J., Mitev, V., Molleker, S., Müller, R., Oelhaf, H., Olschewski, F., Orsolini, Y., Peter, T., Pfeilsticker, K., Piesch, C., Pitts, M. C., Poole, L. R., Pope, F. D., Ravegnani, F., Rex, M., Riese, M., Röckmann, T., Rognerud, B., Roiger, A., Rolf, C., Santee, M. L., Scheibe, M., Schiller, C., Schlager, H., Siciliani de Cumis, M., Sitnikov, N., Søvde, O. A., Spang, R., Spelten, N., Stordal, F., Sumińska-Ebersoldt, O., Ulanovski,
A., Ungermann, J., Viciani, S., Volk, C. M., Vom Scheidt, M., von der Gathen, P., Walker, K., Wegner, T., Weigel, R., Weinbruch, S., Wetzel, G., Wienhold, F. G., Wohltmann, I., Woiwode, W., Young, I. A. K., Yushkov, V., Zobrist, B., and Stroh, F.: Reconciliation of essential process parameters for an enhanced predictability of Arctic stratospheric ozone loss and its climate interactions (RECONCILE): Activities and results, Atmos. Chem. Phys., 13, 9233–9268. https://doi.org/10.5194/acp-13-9233-2013, 2013.

Waibel, A. E., Peter, T., Carslaw, K. S., Oelhaf, H., Wetzel, G., Crutzen, P. J., Pöschl, U., Tsias, A., Reimer, E., and Fischer, H.: Arctic Ozone
Loss Due to Denitrification, Science, 283, 2064–2069. https://doi.org/10.1126/science.283.5410.2064, 1999.

Waters, J. W., Froidevaux, L., Read, W. G., Manney, G. L., Elson, L. S., Flower, D. A., Jarnot, R. F., and Harwood, R. S.: Stratospheric ClO and ozone from the Microwave Limb Sounder on the Upper Atmosphere Research Satellite, Nature, 362, 597. https://doi.org/10.1038/362597a0, 1993.



Waters, J. W., Froidevaux, L., Harwood, R. S., Jarnot, R. F., Pickett, H. M., Read, W. G., Siegel, P. H., Cofield, R. E., Filipiak, M. J., Flower, D. A., Holden, J. R., Lau, G. K., Livesey, N. J., Manney, G. L., Pumphrey, H. C., Santee, M. L., Wu, D. L., Cuddy, D. T., Lay, R. R., Loo, M. S., Perun, V. S., Schwartz, M. J., Stek, P. C., Thurstans, R. P., Boyles, M. A., Chandra, K. M., Chavez, M. C., Chen, G.-S., Chudasama, B. V., Dodge, R., Fuller, R. A., Girard, M. A., Jiang, J. H., Jiang, Y., Knosp, B. W., LaBelle, R. C., Lam, J. C., Lee, K. A., Miller, D., Oswald, J. E., Patel, N. C., Pukala, D. M., Quintero, O., Scaff, D. M., van Snyder, W., Tope, M. C., Wagner, P. A., and Walch, M. J.: The Earth Observing System Microwave Limb Sounder (EOS MLS) on the Aura Satellite, IEEE Trans. Geosc. Remote Sens., 44, 1075–1092. https://doi.org/10.1109/TGRS.2006.873771, 2006.

Wetzel, G., Oelhaf, H., Kirner, O., Friedl-Vallon, F., Ruhnke, R., Ebersoldt, A., Kleinert, A., Maucher, G., Nordmeyer, H., and Orphal, J.: Diurnal variations of reactive chlorine and nitrogen oxides observed by MIPAS-B inside the January 2010 Arctic vortex, Atmos. Chem. Phys., 12, 6581–6592. https://doi.org/10.5194/acp-12-6581-2012, 2012.

Wetzel, G., Oelhaf, H., Birk, M., Lange, A. d., Engel, A., Friedl-Vallon, F., Kirner, O., Kleinert, A., Maucher, G., Nordmeyer, H., Orphal, J., Ruhnke, R., Sinnhuber, B.-M., and Vogt, P.: Partitioning and budget of inorganic and organic chlorine species observed by MIPAS-B and TELIS in the Arctic in March 2011, Atmos. Chem. Phys., 15, 8065–8076. https://doi.org/10.5194/acp-15-8065-2015, 2015.

Winker, D. M., Vaughan, M. A., Omar, A., Hu, Y., Powell, K. A., Liu, Z., Hunt, W. H., and Young, S. A.: Overview of the CALIPSO Mission and CALIOP Data Processing Algorithms, J. Atmos. Ocean. Tech., 26, 2310–2323. https://doi.org/10.1175/2009JTECHA1281.1, 2009.

WMO, ed.: Scientific Assessment of Ozone Depletion: 2006: report of the Montreal Protocol Scientific Assessment Panel, World Meteorological Organization, Geneva. https://library.wmo.int/pmb_ged/gormp_50_en.pdf, 2007.

WMO, ed.: Scientific assessment of Ozone depletion: 2014. Pursuant to Article 6 of the Montreal Protocol on substances that deplete the ozone layer, vol. 55 of *Report / World Meteorological Organization, Global Ozone Research and Monitoring Project*, World Meteorological Organization, Geneva. http://www.wmo.int/pages/prog/arep/gaw/ozone_2014/documents/Full_report_2014_Ozone_Assessment.pdf, 2015.

Woiwode, W., Dörnbrack, A., Bramberger, M., Friedl-Vallon, F., Haenel, F., Höpfner, M., Johansson, S., Kretschmer, E., Krisch, I., Latzko, T., Oelhaf, H., Orphal, J., Preusse, P., Sinnhuber, B.-M., and Ungermann, J.: Mesoscale fine structure of a tropopause fold over mountains, Atmos. Chem. Phys., 18, 15 643–15 667. https://doi.org/10.5194/acp-18-15643-2018, 2018.

Wolff, M. A., Kerzenmacher, T., Strong, K., Walker, K. A., Toohey, M., Dupuy, E., Bernath, P. F., Boone, C. D., Brohede, S., Catoire, V., von Clarmann, T., Coffey, M., Daffer, W. H., Mazière, M. D., Duchatelet, P., Glatthor, N., Griffith, D. W. T., Hannigan, J., Hase, F., Höpfner, M., Huret, N., Jones, N., Jucks, K., Kagawa, A., Kasai, Y., Kramer, I., Küllmann, H., Kuttippurath, J., Mahieu, E., Manney, G., McElroy, C. T., McLinden, C., Mébarki, Y., Mikuteit, S., Murtagh, D., Piccolo, C., Raspollini, P., Ridolfi, M., Ruhnke, R., Santee, M., Senten, C., Smale, D., Tétard, C., Urban, J., and Wood, S.: Validation of HNO3, ClONO2, and N2O5 from the Atmospheric Chemistry Experiment Fourier Transform Spectrometer (ACE-FTS), Atmos. Chem. Phys., 8, 3529–3562. https://doi.org/10.5194/acp-8-3529-2008, 2008.

Zander, R., Rinsland, C. P., Farmer, C. B., Brown, L. R., and Norton, R. H.: Observation of several chlorine nitrate (ClONO2) bands in stratospheric infrared spectra, Geophys. Res. Let., 13, 757–760. https://doi.org/10.1029/GL013i008p00757, 1986.