# Peer review of "Unusual chlorine partitioning in the 2015/16 Arctic winter lowermost stratosphere: Observations and simulations"

_Atmospheric Chemistry and Physics, 2018_

## Referee Comment (RC1) · Anonymous Referee #2 · 20 Mar 2019

<General Comments>

This paper describes the observations of chlorine species (HCl, ClONO2, ClO, and CH3Cl), O3, and HNO3 by satellites (MLS and ACE-FTS) and airborne limb-imager GLORIA, and compared the results with two models (CLaMS and EMAC) in the Arctic winter 2015/16. This winter was characterized by cold stratospheric temperatures, and the study especially focused on the temporal evolutions of relevant trace gases in the lowermost stratosphere (LMS) over the course of the winter. The manuscript is generally well written, well organized, and succeeded to describe some new findings especially related to chlorine deactivation processes in the LMS region. The paper

also describes some problems in the current model simulations which are desired to be improved in near future. I feel that the paper is almost ready to be published in ACP, after some modifications of minor points which are described below.

<Minor Comments/Typos>

1) P.6, L.1: CALIPSO satellite was launched 2006 –> CALIPSO satellite was launched in 2006

2) P.6, L.6: McKenna et al., 2002b, a –> McKenna et al., 2002a, b

3) P.6, L.8: initialized 1 November 2015 –> initialized on 1 November 2015

4) P.6, L.18: comparison of CLaMS results to GLORIA and MLS –> comparison of CLaMS results with GLORIA and MLS

5) P.7, L.1: analyses has been initialized 1 July 2015 –> analyses has been initialized on 1 July 2015

6) P.17, Figure 8 caption: Colorbars –> Color bars

7) P.17, Figure 8 caption: colorbar –> color bar

8) P.24, L.15: Fig. 12a1 –> Fig. 12(a1)

9) P.24, L.15: Fig. 12b1 –> Fig. 12(b1)

10) P.27, L.17: Fig. 12b2 –> Fig. 12(b2)

11) P.27, L.19: Fig. 12e2 –> Fig. 12(e2)

12) P.28, L.11: which is nearly all ClO at this time –> which consists of ClO and $Cl_2O_2$ at this time

Please also note the supplement to this comment:
https://www.atmos-chem-phys-discuss.net/acp-2018-1227/acp-2018-1227-RC1-supplement.pdf

---

## Referee Comment (RC2) · Anonymous Referee #1 · 21 Mar 2019

This work examines the heterogeneous chemistry in the 2015/2016 Arctic winter, LMS, during the POLSTRACC/GW-LCYCLEII/GWEX/SALSA campaigns. The GLORIA instrument aboard aircraft and the satellite instrument Aura MLS was used to examine activation and deactivation of inorganic chlorine, ozone depletion, and irreversible denitrification. Processes were evaluated using both the state-of-the-art lagrangian chemical transport model (CLaMS) and Eulerian (EMAC) CTM. This is an excellent study that show how observations taken on different platforms (i.e., satellite and aircraft), coupled with model analysis can explain and document the chemical processes in the atmosphere. This paper is appropriate for ACP. I have a few comments below that I would recommend the authors to consider.

[Figure]

Title: Is this winter that "unusual"? I.e., when the temperatures get cold enough (in the Arctic), the atmosphere will be denitrified, heterogeneous processes happen, ozone will be depleted (and if low enough), will recovery into HCl, not ClONO2. You do make some nice points about the role of O3 depletion and denitrification, e.g., you say "The (for the Arctic) unusual chlorine deactivation has been identified through CLaMS sensitivity studies to result at 380 K from low O3 abundances rather than from low NOy availability caused by PSC sedimentation. At higher potential temperatures (as shown at 490 K), denitrification played a greater role." So, my point is this, is the 2015/16 Arctic winter "unusual", or is it that this winter happened to have multiple satellite instruments flying, along with an impressive field campaign, with mature chemical models to confirm what is already known?

Introduction: Portmann et al., JGR, 1996 (ClONO2 reference) should be added. This paper was one of the first papers that gave detailed explanation on the polar chemistry of ClONO2 and how denitrification plays a role.

Page 6. Is there a reference for the STS parameterization in CLaMS? Is this based on Carslaw et al. 1994?

Page 7. Section 2.3.2, EMAC. I realize that the PSC representation is discussed in Khosrawi et al., 2017, but for the reader it would be nice to have several sentences that discuss the PSC representation in EMAC.

General Comment on comparing to Satellite observations. There are a lot of detail comments about how well CLaMS compares to MLS and ACE-FTS. Frankly, I am very impressed with the comparisons. However, in the LMS, where there are gradients and where there are differences from the observations and model – can this not be, at least partially, due to the 3-4km retrieved vertical resolution of the observations (especially when the stated vertical resolution of the model in this region is ∼800m)? I assume you have not applied the averaging kernel from the observations to the model results? If not, it would be interesting to discuss the impact of NOT including the AVK when

discussing Figure 3 and 4.

Comment on comparison of CLaMS and EMAC to observation in Figure 8. The CLaMS model representation of the GLORIA observations is excellent. The EMAC, model, even at 1-degree horizontal resolution seem to have issues representing ozone (Figure 8) and ClONO2 (Figure 9). On page 19, the authors make several good comments on the differences between CLaMS and EMAC that may contribute to EMAC not representing either CLaMS or GLORIA results. The authors suggest that EMAC has issues with representing downward transport in the lower part of the polar vortex. Besides transport, is there any reason to believe that there are chemistry differences between the two model frameworks that could be contributing to the differences. E.g., are the heterogeneous chemistry modules similar between EMAC and CLaMS? Some discussion here would be useful. In addition, it is not clear to me why the EMAC model results are in this paper?? The CLaMS model is useful since it represents the observations very well and has been used to understand chemical processes. What is the role of the EMAC model?

Figure 10. Panels (i) and (j) are unreadable if you print out the paper. One can view your symbols if you blow up the PDF on a large monitor. I would suggest move the caption (below panel "j") and expand the panels, making them larger.

Definition of ozone loss tracer. Does it make a difference if the ozone loss tracer includes gas phase chemistry? I believe you are assuming that there is no chemistry included in this tracer, correct?

---

## Author Comment (AC1) · 8 May 2019

We thank referee 2 for valuable comments and suggestions. Our answers are given below. The original referee comment is repeated in **bold**, changes in the manuscript text are printed in *italic*.

<**Minor Comments/Typos**>

**1) P.6, L.1: CALIPSO satellite was launched 2006 → CALIPSO satellite was launched in 2006**

[Figure]

We changed the manuscript according to the referee's suggestion.

**2) P.6, L.6: McKenna et al., 2002b, a → McKenna et al., 2002a, b**
We changed the manuscript according to the referee's suggestion.

**3) P.6, L.8: initialized 1 November 2015 → initialized on 1 November 2015**
We changed the manuscript according to the referee's suggestion.

**4) P.6, L.18: comparison of CLaMS results to GLORIA and MLS → comparison of CLaMS results with GLORIA and MLS**
We changed the manuscript according to the referee's suggestion.

**5) P.7, L.1: analyses has been initialized 1 July 2015 → analyses has been initialized on 1 July 2015**
We changed the manuscript according to the referee's suggestion.

**6) P.17, Figure 8 caption: Colorbars → Color bars**
**7) P.17, Figure 8 caption: colorbar → color bar**
We changed the spelling according to the referee's suggestion.

**8) P.24, L.15: Fig. 12a1 → Fig. 12(a1)**
**9) P.24, L.15: Fig. 12b1 → Fig. 12(b1)**
**10) P.27, L.17: Fig. 12b2 → Fig. 12(b2)**
**11) P.27, L.19: Fig. 12e2 → Fig. 12(e2)**
We changed the format of the figure references according to the referee's suggestion.

**12) P.28, L.11: which is nearly all ClO at this time → which consists of ClO and Cl2O2 at this time**
We changed the indicated text to: *which consists mainly of ClO, with small contribu-*

*tions from Cl$_2$O$_2$ at this time.* We think it is important to stress that the main part of ClO$_x$ is ClO and only minor parts of ClO$_x$ along this trajectory is Cl$_2$O$_2$.

---

## Author Comment (AC2) · 8 May 2019

We thank referee 1 for valuable comments and suggestions. Our answers are given below. The original referee comment is repeated in **bold**, changes in the manuscript text are printed in *italic*.

**Title: Is this winter that "unusual"? I.e., when the temperatures get cold enough (in the Arctic), the atmosphere will be denitrified, heterogeneous processes happen, ozone will be depleted (and if low enough), will recovery into HCl, not ClONO2. You do make some nice points about the role of O3**

[Figure]

**depletion and denitrification, e.g., you say "The (for the Arctic) unusual chlorine deactivation has been identified through CLaMS sensitivity studies to result at 380 K from low O3 abundances rather than from low NOy availability caused by PSC sedimentation. At higher potential temperatures (as shown at 490 K), denitrification played a greater role." So, my point is this, is the 2015/16 Arctic winter "unusual", or is it that this winter happened to have multiple satellite instruments flying, along with an impressive field campaign, with mature chemical models to confirm what is already known?**

We think that the Arctic winter 2015/16 was indeed unusual regarding chlorine partitioning compared to any other winter during the Aura/MLS epoch. In particular, we show that HCl in January and February at a potential temperature of 380 K has never been measured as low as in 2015/16 in the time between 2004 and 2018 (see Fig. 2). These low HCl measurements have been observed even though other winters have shown lower temperatures at these potential temperature levels. In our opinion this substantial deviation from climatological behavior should justify to call the chlorine partitioning "unusual" for this Arctic winter. But, in fact, our measurements during the PGS field campaign did draw our attention to the lowermost stratosphere in available satellite data during that winter. In order to strengthen the focus on the unusual aspect of the 2015/16 winter, we changed the conclusions on P29/L7 to: *The analysis of ACE-FTS and MLS time series shows the unusual nature of the Arctic winter 2015/16.*

**Introduction: Portmann et al., JGR, 1996 (ClONO2 reference) should be added. This paper was one of the first papers that gave detailed explanation on the polar chemistry of ClONO2 and how denitrification plays a role.**

Thank you for pointing out this reference, which we included in the introductory section at P2/L15 (the reference section was updated accordingly).

**Page 6. Is there a reference for the STS parameterization in CLaMS? Is this based on Carslaw et al. 1994?**

Thank you for pointing that out! In fact, there are other helpful references describing the STS parameterization in CLaMS, which we add to the model description part at P6/L27 (the reference section was updated accordingly):

Carslaw, K. S., Clegg, S. L., and Brimblecombe, P.: A Thermodynamic Model of the System HCl-HNO3-H2SO4-H2O, Including Solubilities of HBr, from <200 to 328 K, J. Phys. Chem., 99, 11557–11574, doi:10.1021/j100029a039, 1995.

Carslaw, K. S., Luo, B., and Peter, T.: An analytic expression for the composition of aqueous HNO3-H2SO4 stratospheric aerosols including gas phase removal of HNO3, Geophys. Res. Lett., 22, 1877–1880, doi:10.1029/95GL01668, 1995.

**Page 7. Section 2.3.2, EMAC. I realize that the PSC representation is discussed in Khosrawi et al., 2017, but for the reader it would be nice to have several sentences that discuss the PSC representation in EMAC.**

We added to the model description the following text at P7/L5 (new references were added to the reference section accordingly): *The EMAC "Multi-phase Stratospheric Box Model" module simulates the number densities, mean radii and surface areas of sulfuric acid aerosols and liquid and solid PSC particles. The formation of STS particles is calculated according to Carslaw (2002). Ice particles are assumed to form homogeneously at temperatures below $T_{ice}$ and the sedimentation of these particles is calculated according to Waibel et al. (1999). NAT formation is calculated using the "kinetic NAT parameterization" which is based on the growth and sedimentation algorithm given by Carslaw (2002) and van den Broek et al. (2004).*

Carslaw, K. S., Luo, B., and Peter, T.: An analytic expression for the composition of aqueous HNO3-H2SO4 stratospheric aerosols including gas phase removal of HNO3, Geophys. Res. Lett., 22, 1877–1880, doi:10.1029/95GL01668, 1995.

Broek, M. M. P. van den, Williams, J. E., and Bregman, A.: Implementing growth and sedimentation of NAT particles in a global Eulerian model, Atmospheric Chemistry and Physics, 4, 1869–1883, doi:10.5194/acp-4-1869-2004, 2004.

Waibel, A. E., Peter, T., Carslaw, K. S., Oelhaf, H., Wetzel, G., Crutzen, P. J., Pöschl,

U., Tsias, A., Reimer, E., and Fischer, H.: Arctic Ozone Loss Due to Denitrification, Science, 283, 2064–2069, doi:10.1126/science.283.5410.2064, 1999.

**General Comment on comparing to Satellite observations. There are a lot of detail comments about how well CLaMS compares to MLS and ACE-FTS. Frankly, I am very impressed with the comparisons. However, in the LMS, where there are gradients and where there are differences from the observations and model – can this not be, at least partially, due to the 3-4km retrieved vertical resolution of the observations (especially when the stated vertical resolution of the model in this region is ≈800m)? I assume you have not applied the averaging kernel from the observations to the model results? If not, it would be interesting to discuss the impact of NOT including the AVK when discussing Figure 3 and 4.**

You are right that we did not apply the MLS averaging kernels to the CLaMS model results. As a compromise between computational effort and precise comparison, we decided to average the CLaMS model on isentropic levels at the geolocations of the MLS measurements without the application of the averaging kernels to the Lagrangian model. In Fig. 1 of this answer, we show for an exemplary $HNO_3$ profile the difference between daily vortex average profiles with (blue) and without (green) the application of a typical MLS averaging kernel. For this example, the difference of the profiles with and without averaging kernel application (see Fig. 1 of this answer, right panel) is considerably lower than the differences between measurement and model discussed in the paper. For $HNO_3$, the exemplary error due to the non-application of the MLS averaging kernel is 0.2 ppbv, while differences of 1.0 ppbv between measurement and model are discussed in our paper. We added to the manuscript at P10/L17: *An exemplary comparison of CLaMS $HNO_3$ daily vortex average profiles with and without the application of typical MLS averaging kernels showed considerably smaller differences than the differences between measurement and model that are discussed in this work. Due to this minor impact, CLaMS data is shown without the application of*

*MLS averaging kernels.*

**Comment on comparison of CLaMS and EMAC to observation in Figure 8. The CLaMS model representation of the GLORIA observations is excellent. The EMAC, model, even at 1-degree horizontal resolution seem to have issues representing ozone (Figure 8) and ClONO2 (Figure 9). On page 19, the authors make several good comments on the differences between CLaMS and EMAC that may contribute to EMAC not representing either CLaMS or GLORIA results. The authors suggest that EMAC has issues with representing downward transport in the lower part of the polar vortex. Besides transport, is there any reason to believe that there are chemistry differences between the two model frameworks that could be contributing to the differences. E.g., are the heterogeneous chemistry modules similar between EMAC and CLaMS? Some discussion here would be useful. In addition, it is not clear to me why the EMAC model results are in this paper?? The CLaMS model is useful since it represents the observations very well and has been used to understand chemical processes. What is the role of the EMAC model?**

The chemistry schemes of CLaMS and EMAC both include all relevant heterogeneous and gas phase reactions for the polar winter stratosphere and both are based on Carslaw et al. (1995a, 1995b). We added to our manuscript at P19/L4 : *The differences in the models' representations of measured trace gas distributions are expected to result from resolution and dynamics rather than from the modeled chemistry, as both models are based on the same chemistry scheme (see Sec. 2.3).*

We included EMAC in the comparison since as discussed in Khosrawi et al. (2017), we can show with these applications that EMAC simulations nudged toward ECMWF operational analysis can reproduce the observations within the limitations of the applied model resolution and process parameterizations. In Khosrawi et al. (2017) this is shown for ozone and nitric acid, while here additionally the same is shown for $ClONO_2$. Therefore, the comparison of EMAC to GLORIA shows that, though

EMAC is a chemistry-climate model, EMAC simulations can be applied in support of aircraft campaigns and as a valuable data set not only for flight analyses but also for process studies and realistic future projections. Vice-versa, as we show that the EMAC implementation of stratospheric polar ozone chemistry is very probably correct, its application in EMAC when run as climate model is evaluated. We added to the manuscript on P20/L7: *Nevertheless, these comparisons confirm the results of Khosrawi et al. 2017 that EMAC, though a chemistry-climate model, can be applied in support of aircraft campaigns and as a valuable data set not only for flight analyses but also for process studies and realistic future projections.*

In addition, we changed the conclusions on P29/L21 to: *In addition, well-known problems of EMAC's diabatic descent are observed in the comparisons, but generally it is shown that EMAC can support aircraft campaigns for process studies and realistic future projections.*

Carslaw, K. S., Clegg, S. L., and Brimblecombe, P.: A Thermodynamic Model of the System HCl-HNO3-H2SO4-H2O, Including Solubilities of HBr, from <200 to 328 K, J. Phys. Chem., 99, 11557–11574, doi:10.1021/j100029a039, 1995a.

Carslaw, K. S., Luo, B., and Peter, T.: An analytic expression for the composition of aqueous HNO3-H2SO4 stratospheric aerosols including gas phase removal of HNO3, Geophys. Res. Lett., 22, 1877–1880, doi:10.1029/95GL01668, 1995b.

Khosrawi, F., Kirner, O., Sinnhuber, B.-M., Johansson, S., Höpfner, M., Santee, M. L., Froidevaux, L., Ungermann, J., Ruhnke, R., Woiwode, W., Oelhaf, H., and Braesicke, P.: Denitrification, dehydration and ozone loss during the 2015/2016 Arctic winter, Atmospheric Chemistry and Physics, 17, 12893–12910, doi:10.5194/acp-17-12893-2017, 2017.

**Figure 10. Panels (i) and (j) are unreadable if you print out the paper. One can view your symbols if you blow up the PDF on a large monitor. I would suggest move the caption (below panel "j") and expand the panels, making them larger.**

We improved the readability of Fig. 10 by increasing the vertical size of the figure and by using larger symbols for plotting panels (a), (b), (i), and (j).

**Definition of ozone loss tracer. Does it make a difference if the ozone loss tracer includes gas phase chemistry? I believe you are assuming that there is no chemistry included in this tracer, correct?**

Our "passive ozone" tracer is only transported and does not experience any chemical reaction. Therefore, the ozone loss which is presented in our paper shows the ozone loss caused by heterogeneous and gas phase chemistry. Of course, this ozone loss tracer does not solely show the effect of activated chlorine on ozone loss. However, in the lowermost stratosphere, ozone loss is expected to result mostly from heterogeneous activated chlorine rather than other gas phase ozone loss cycles.

Introducing an additional "semi-passive ozone" tracer, which experiences gas phase chemistry but neglecting heterogeneous chlorine activation, would of course exclusively show the influence of heterogeneous reactions on ozone depletion, but this additional tracer would also stretch our already quite extensive study. For that reason, we prefer to only discuss the purely passive ozone tracer in our work.

For clarification, we added to the manuscript at P20/L30 : *Due to the definition of the passive $O_3$ tracer, this presented ozone loss may be caused by both, heterogeneous and gas phase, chemical reaction types. According to Singleton et al. (2005), mostly heterogeneous reactions are responsible for ozone depletion.*

Singleton, C. S., Randall, C. E., Chipperfield, M. P., Davies, S., Feng, W., Bevilacqua, R. M., Hoppel, K. W., Fromm, M. D., Manney, G. L., and Harvey, V. L.: 2002-2003 Arctic ozone loss deduced from POAM III satellite observations and the SLIM-CAT chemical transport model, Atmospheric Chemistry and Physics, 5, 597–609, doi:10.5194/acp-5-597-2005, 2005.
* * *
[Figure]

**Fig. 1.** Left: Exemplary vortex mean profiles (HNO3 on 26 February 2016) from CLaMS with (blue) and without (green) the application of the MLS averaging kernel. Right: Difference of profiles with & without AVK

---

## Author Response (AR2)

**Answer to Co-Editor Comment**

**May 27, 2019**

We thank Mathias Palm for valuable comments and suggestions. Our answers are given below. The original comment is repeated in **bold**, changes in the manuscript text are printed in *italic*.

**1. Please put a legend on figure 1. I am not sure if people which are not from Europe will immediately see, where you operated. The same is true for panel a in figures 5 to 7 and subpanels h and g in figure 11. I know that this is more or less double information, but in my view it would make it easier to read and to concentrate on the contents of the paper rather than geography.**

The legend on Figure 1 assigns different on the plot to the corresponding flight dates. Instead of adjusting the legend, we added to the Figure caption: *The underlain map (bold black lines show coastal lines, regular black lines country borders, dotted lines indicate a latitude/longitude grid) shows the European continent and is centered at $60^\circ$ N latitude / $10^\circ$ E longitude.* For Figures 5-7,11 we added similar statements.

**2. In figure 8 you explain what you mean with arrows at the colorbars, but you seems to use them in figures before too. Or is this a misconception?**

Thank you for pointing that out! We added to the captions of Figures 3 and 5 the sentence: *Color bars with top/bottom arrowheads indicate data greater/smaller than the color bar range.* The captions of Figures 6 and 7 only refer to the caption of Figure 5 and we therefore did not repeat this sentence again.

**3. The abstract and the conclusion are often read first to assess if**

**the manuscript is interesting (for the particular reader not in General). It therefor enhance readability if the abstract and conclusion can be read independently from the main text. In order to do so, I suggest:**
**You use the abbreviation LMS with expanding it again. LWS seems to be a rather unusual term, so it would enhance readability to mention it again. Also, please write measurements from the GLORIA instrument (instead of measurements from GLORIA).**

We removed the abbreviation "LMS" from the abstract and redefined it in the beginning of the conclusions. We also changed the "measurements from GLORIA" part according to your suggestion.

**In line 12 of page 27 you write '..Although ozone loss was greater in 2010/2011, 2015/16 was a winter with extraordinary large chemical ozone loss. ...' This left me wondering if there are other 'types of ozone loss' and what they are. I think most people would equate 'ozone loss' with 'chemical ozone loss'.**

According to your suggestion, we removed the word "chemical".

[revised manuscript text omitted]